# C-LoRA: Contextual Low-Rank Adaptation for Uncertainty Estimation in Large Language Models

**Amir Hossein Rahmati**[1]    **Sanket Jantre**[2]    **Weifeng Zhang**[1]    **Yucheng Wang**[1]
**Byung-Jun Yoon**[1,2]    **Nathan M. Urban**[2]    **Xiaoning Qian**[1,2]

[1]Texas A&M University, College Station, TX    [2]Brookhaven National Laboratory, Upton, NY

{amir_hossein_rahmati, weifengzhang, wangyucheng, bjyoon, xqian}@tamu.edu
{sjantre, nurban, byoon, xqian1}@bnl.gov

## Abstract

Low-Rank Adaptation (LoRA) offers a cost-effective solution for fine-tuning large language models (LLMs), but it often produces overconfident predictions in data-scarce few-shot settings. To address this issue, several classical statistical learning approaches have been repurposed for scalable uncertainty-aware LoRA fine-tuning. However, these approaches neglect how input characteristics affect the predictive uncertainty estimates. To address this limitation, we propose Contextual Low-Rank Adaptation (**C-LoRA**) as a novel uncertainty-aware and parameter efficient fine-tuning approach, by developing new lightweight LoRA modules contextualized to each input data sample to dynamically adapt uncertainty estimates. Incorporating data-driven contexts into the parameter posteriors, C-LoRA mitigates overfitting, achieves well-calibrated uncertainties, and yields robust predictions. Extensive experiments on LLaMA2-7B models demonstrate that C-LoRA consistently outperforms the state-of-the-art uncertainty-aware LoRA methods in both uncertainty quantification and model generalization. Ablation studies further confirm the critical role of our contextual modules in capturing sample-specific uncertainties. C-LoRA sets a new standard for robust, uncertainty-aware LLM fine-tuning in few-shot regimes. Although our experiments are limited to 7B models, our method is architecture-agnostic and, in principle, applies beyond this scale; studying its scaling to larger models remains an open problem. Our code is available at https://github.com/ahra99/c_lora.

## 1   Introduction

Large Language Models (LLMs) [1–7] have shown their promising potential in diverse areas [8–18]. Due to their general-purpose language understanding and generation capabilities with human-level performance [1, 2], fine-tuning LLMs to various downstream tasks has drawn significant attention [19–23]. However, when fine-tuned for downstream tasks with limited data, LLMs may hallucinate to produce overconfident results, which becomes a serious concern [24–27]. To address this, reliable estimation of uncertainty has become essential [19, 20, 28].

To enable predictive uncertainty quantification (UQ), probabilistic inference with Bayesian neural network (BNN) has been studied in deep learning [29–41], where neural network weights are treated as random variables with variational inference (VI) approximating the true posterior to provide reliable UQ [42, 43]. Adopting these approaches for LLMs is limited by their prohibitive computational and memory costs compared to conventional methods [19, 20, 22], making full Bayesian fine-tuning of all the parameters of an LLM challenging.

39th Conference on Neural Information Processing Systems (NeurIPS 2025).

Parameter Efficient Fine-Tuning (PEFT) methods, such as Low-rank Adaptation (LoRA), substantially reduce the number of learnable parameters and thereby mitigate the significant computational expenses and excessive memory utilization [21, 44–48]. This efficiency facilitates employing Bayesian methods for UQ in LLMs [19, 20, 49–51]. Most of the recent studies considered a more straightforward solution, like in [49–51] where authors considered ensemble methods. [19] proposed a *post-hoc* Bayesian inference method for the LoRA adapters using Laplace approximation [34]. [20] repurposed a mean-field VI based Bayesian framework for LoRA-based fine-tuning to jointly estimate the LoRA parameters' variational means and variances [20]. However, the uncertainty stemming from data, *aleatoric uncertainty*, has not been considered in the existing methods, leading to poor performance especially when fine-tuning with limited data. This motivates us to focus on incorporating aleatoric uncertainty for a novel parameter efficient fine-tuning approach in this work.

We propose **C**ontextual **L**ow-**R**ank **A**daptation, **C-LoRA**, which enables an explicit consideration of aleatoric uncertainty (data uncertainty). This formulation not only leads to faster training, but also provides a sample-dependent uncertainty estimation for each layer, which leads to beneficial and more favorable uncertainty-aware LLM fine-tuning under small-data scenarios. In particular, we introduce a contextual module for modeling the stochasticity in low-dimensional space dependent on the data, which enables a low-cost contextualized estimation of prediction uncertainty in terms of computation and memory usage. Our contributions can be outlined as follows:

- We propose a new end-to-end Bayesian framework for scalable uncertainty-aware LLM fine-tuning via contextualized LoRA on data in the lower-dimensional space by a flexible contextual module;

- Our framework allows for efficient modeling of aleatoric (data) uncertainty;

- We showcase the superiority of C-LoRA regarding both UQ capabilities and generalizability via extensive experiments across different tasks;

- Through ablation experiments, we demonstrate the significance of our new contextual module on achieving better UQ while offering competitive accuracy with only minor drops on some tasks.

## 2 Preliminaries

In this paper, vectors and matrices are denoted by bold lowercase and uppercase letters, respectively. Most of our mathematical notations follow the ones adopted in [21], [19], [20], and [39].

### 2.1 Low-Rank Adaptation (LoRA)

LoRA, a parameter-efficient fine-tuning approach, adapts a pre-trained language model to downstream tasks [21]. It rests on the assumption that the required weight updates have a low intrinsic dimensionality, so LoRA freezes the original weights and instead learns low-rank update matrices. To this end, the modified forward pass becomes:

$$\mathbf{h} = (\mathbf{W}_0 + \Delta\mathbf{W})\mathbf{x} = (\mathbf{W}_0 + \mathbf{B}\mathbf{A})\mathbf{x}, \tag{1}$$

where $\mathbf{x} \in \mathbb{R}^k$ and $\mathbf{h} \in \mathbb{R}^d$ are input and output vectors, respectively, and $\mathbf{W}_0 \in \mathbb{R}^{d \times k}$ represents the frozen pre-trained weights. LoRA inserts two low-rank update factors $\mathbf{B} \in \mathbb{R}^{d \times r}$ and $\mathbf{A} \in \mathbb{R}^{r \times k}$ with $r \ll \min(d, k)$. Hence, the number of trainable parameters reduces to $r \times (d + k)$ which is considerably lower than $d \times k$ in the full matrix. This dramatically cuts storage and computational costs while matching full-matrix fine-tuning performance. Here on, we set $k = d$, so that $\mathbf{W}_0 \in \mathbb{R}^{d \times d}$.

### 2.2 Bayesian Uncertainty Estimation

Let $\mathcal{D} = \{\mathbf{x}_i, y_i\}_{i=1}^N$ be a dataset of $N$ independent and identically distributed (i.i.d.) observations, where each $\mathbf{x}_i$ is an input sample and $y_i$ is the corresponding output. In the Bayesian paradigm, rather than selecting a single best-fit model parameterization, we maintain a distribution over all plausible model parameters. Specifically, given model parameters $\boldsymbol{\theta}$, Bayesian inference aims to characterize the posterior distribution $p(\boldsymbol{\theta}|\mathcal{D})$ using Bayes' rule: $p(\boldsymbol{\theta}|\mathcal{D}) \propto p(\mathcal{D}|\boldsymbol{\theta}) p(\boldsymbol{\theta})$, where $p(\mathcal{D}|\boldsymbol{\theta})$ is the likelihood of the observed data under parameters $\boldsymbol{\theta}$, and $p(\boldsymbol{\theta})$ is the prior distribution reflecting beliefs about $\boldsymbol{\theta}$ before seeing any data.

To generate predictions for a new input $\mathbf{x}^*$, Bayesian model averaging, which integrates over the posterior is applied and the intractable integral is approximated using Monte Carlo sampling:

$$p(y^*|\mathbf{x}^*, \mathcal{D}) = \int p(y^*|\mathbf{x}^*, \boldsymbol{\theta}) \, p(\boldsymbol{\theta}|\mathcal{D}) \, d\boldsymbol{\theta} \approx \frac{1}{M} \sum_{m=1}^{M} p(y^*|\mathbf{x}^*, \boldsymbol{\theta}_m), \quad \boldsymbol{\theta}_m \sim p(\boldsymbol{\theta}|\mathcal{D}). \tag{2}$$

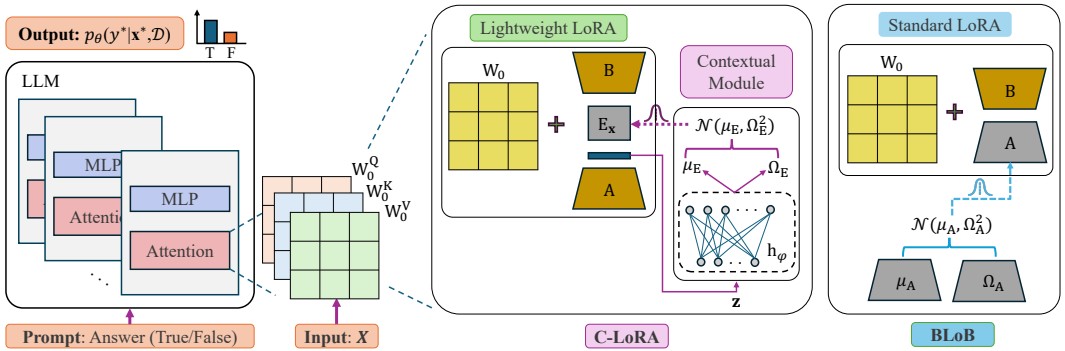

Figure 1: A visual representation of our proposed method, Conextual LoRA **(C-LoRA)** and the Bayesian LoRA by backpropagation **(BLoB)**.

## 2.3 Bayesian Low-Rank Adaptation

Despite the existence of scalable posterior inference methods, a full Bayesian treatment of LLMs remains computationally prohibitive. By restricting Bayesian updates to the LoRA parameters, a tractable uncertainty quantification scheme can be achieved; however, even Markov chain Monte Carlo (MCMC) over millions of LoRA weights is too costly. As a practical compromise, *Bayesian LoRA by backpropagation (BLoB)* [20] embeds uncertainty estimation directly into fine-tuning via mean-field variational inference on LoRA adapters. More specifically, they keep $\mathbf{B}$ deterministic in (1) and Bayesianize $\mathbf{A}$ with a variational distribution $q(\mathbf{A}) = \mathcal{N}(\mathbf{A}|\boldsymbol{\mu}_{\mathbf{A}}, \boldsymbol{\Omega}_{\mathbf{A}}^{2})$, where $\boldsymbol{\mu}_{\mathbf{A}}$ and $\boldsymbol{\Omega}_{\mathbf{A}}^{2}$ are the variational mean and variance estimates, respectively. To this end, they learn the variational distribution parameters by maximizing the Evidence Lower BOund (ELBO):

$$\mathcal{L}' = \mathbb{E}_q\big[\log p(\mathcal{D}|\mathbf{A}, \mathbf{B})\big] \ - \ \mathrm{KL}\big[q(\mathbf{A}) \,\|\, p(\mathbf{A})\big]. \tag{3}$$

Here, the first term measures the expected negative log-likelihood of the data under the variational posterior, while the second term is the Kullback-Leibler (KL) divergence between the variational posterior and the prior, acting as a regularization. Using the reparameterization trick, BLoB jointly updates means and variances providing scalable, predictive uncertainty estimates. A visual representation of BLoB framework is presented in Figure 1.

## 3 Contextual Low-Rank Adaptation (C-LoRA)

We introduce the formulation of Contextual Low-Rank Adaptation (C-LoRA), which enables scalable, efficient, data-dependent uncertainty quantification at the sample level. First, we introduce a lightweight LoRA factorization to reduce the computational burden of variational inference in Bayesian LoRA. Based on this factorization, we treat the LoRA weights' stochasticity to be data-dependent, and learn the weight distribution parameters via a variational Bayesian objective. A step-by-step description of C-LoRA is presented in Algorithm 1 (Appendix F).

### 3.1 Lightweight LoRA Factorization

The standard LoRA update in (1) introduces two low-rank matrices whose size scales with the frozen weight dimension $d$, so that any Bayesian treatment—whether both adapters are stochastic or only $\mathbf{A}$ is (as in BLoB)—incurs a computational cost that grows linearly in $d$. To break this dependency, we insert a low-dimensional matrix $\mathbf{E} \in \mathbb{R}^{r \times r}$ between $\mathbf{B}$ and $\mathbf{A}$. This modification reduces the stochastic parameter complexity to a constant in $d$, yielding a modified LoRA factorization as follows

$$\mathbf{h} = (\mathbf{W}_0 + \Delta\mathbf{W})\mathbf{x} = (\mathbf{W}_0 + \mathbf{BEA})\mathbf{x}. \tag{4}$$

Compared to the BLoB mean-field VI framework, this new factorization facilitates a more scalable, lightweight Bayesian LoRA by inferring a distribution over the elements of $\mathbf{E}$ while learning $\mathbf{B}$ and $\mathbf{A}$ deterministically, making data-dependent Bayesianization of LoRA fine-tuning scalable.

## 3.2 Stochastic LoRA Parameterization with Data Dependence

In conventional Bayesian neural networks, the focus is on epistemic uncertainties introduced by treating model parameters as random variables drawn from a fixed distribution—one that does not depend on the input data. Consequently, while the sampled parameters may vary across data instances, their underlying distribution remains invariant across all samples in the training set. This assumption limits the expressiveness of uncertainty estimates, particularly in few-shot LoRA fine-tuning scenarios. To address this, we propose a data-dependent, or *contextual*, Bayesian fine-tuning paradigm, wherein the distribution of the parameters of low-rank adapters depends on the inputs $\mathbf{x}_i$ for each data sample $(\mathbf{x}_i, y_i)$. Although one could perform Bayesian inference over $\mathbf{B}$ and $\mathbf{A}$ LoRA adapters from (1), this scales linearly with the frozen weight dimension $d$. Instead, by leveraging the lightweight LoRA factorization from (4), we learn the low-dimensional $\mathbf{E}$ matrix contextually, yielding the weight updates as

$$\Delta \mathbf{W} = \mathbf{B} \mathbf{E}_{\mathbf{x}} \mathbf{A} \tag{5}$$

where $\mathbf{E}_{\mathbf{x}}$ denotes the data-dependent version of $\mathbf{E}$. To be specific, we take an input-dependent Gaussian variational posterior over $\mathbf{E}_{\mathbf{x}_i}$, $q_\phi(\mathbf{E}_{\mathbf{x}_i}|\mathbf{x}_i) = \mathcal{N}(\boldsymbol{\mu}_{\mathbf{E}}(\mathbf{x}_i), \boldsymbol{\Omega}_{\mathbf{E}}^2(\mathbf{x}_i))$. We learn these distribution parameters via lightweight per-layer auxiliary contextual modules comprising of small neural networks whose parameters across modules are collectively denoted as $\boldsymbol{\varphi}$. Such an input-dependent variational posterior resembles amortized inference in Bayesian modeling where $q_\phi$ serves as an inference network that approximates $p(\mathbf{E}_{\mathbf{x}_i}|y_i, \mathbf{x}_i) \propto p(y_i|\mathbf{x}_i, \mathbf{E}_{\mathbf{x}_i})p(\mathbf{E}_{\mathbf{x}_i})$.

Particularly, consider a pre-trained LLM with $L$ layers, each augmented by its own LoRA adapters $\mathbf{B}^l$, $\mathbf{E}_{\mathbf{x}}^l$, and $\mathbf{A}^l$. Let $\mathbf{x}_i^{l-1}$ denote the output of layer $l-1$. We then model $\mathbf{E}_{\mathbf{x}_i}^l$ autoregressively by conditioning on $\{\mathbf{E}_{\mathbf{x}_i}^j\}_{j<l}$, leading to $q_\phi(\mathbf{E}_{\mathbf{x}_i}|\mathbf{x}_i) = \prod_{l=1}^L q_\phi(\mathbf{E}_{\mathbf{x}_i}^l|\mathbf{x}_i^{l-1})$. Especially, in layer $l$, given $\mathbf{z}^l = \mathbf{A}^l \mathbf{x}^{l-1} \in \mathbb{R}^r$ as input, the corresponding contextual module denoted by $h_\varphi^l$ produces the parameters $(\boldsymbol{\mu}_{\mathbf{E}}^l, \boldsymbol{\Omega}_{\mathbf{E}}^l)$ of the $\mathbf{E}_{\mathbf{x}}^l$ distribution as output. We then draw $\mathbf{E}_{\mathbf{x}}^l$ conditioned on $\boldsymbol{\mu}_{\mathbf{E}}^l$ and $\boldsymbol{\Omega}_{\mathbf{E}}^l$. Finally, multiplying $\mathbf{B}^l$, $\mathbf{E}_{\mathbf{x}}^l$, and $\mathbf{z}^l$ and adding it to $\mathbf{W}_0^l \mathbf{x}^{l-1}$ yields the output of the layer $l$, $\mathbf{x}^l$.

This formulation enables us to focus exclusively on *heteroscedastic uncertainty*—modeling the variability in $y_i$ given $\mathbf{x}_i$—with the epistemic uncertainty modeling available via imposing prior on $\boldsymbol{\varphi}$ or Bayesianizing $\mathbf{A}$ or $\mathbf{B}$ when needed, thereby isolating and highlighting the advantages of data-dependent (heterscedetic) UQ.

**Contextual module parameterization.** We parameterize each layer's contextual module with a two fully-connected layer neural network whose parameters at layer $l$ are denoted as $\varphi^l$. In particular, each of these neural networks have $r$ inputs, $C$ hidden units, and $2 \times r^2$ outputs with the nonlinear ReLU activation function connecting the two fully-connected layers.

### 3.3 Amortized Inference for Contextual LoRA

In our formulation, the variational distribution is only conditioned on $\mathbf{x}_i$, while $y_i$ contributes through the training objective. We then learn the model parameters, $\phi$, by maximizing the ELBO of $\sum_i \log p(y_i|\mathbf{x}_i) = \sum_i \log \int p(y_i|\mathbf{x}_i, \mathbf{E}_{\mathbf{x}_i})p(\mathbf{E}_{\mathbf{x}_i})d\mathbf{E}_{\mathbf{x}_i}$ given $q_\phi(\mathbf{E}_{\mathbf{x}_i}|\mathbf{x}_i)$:

$$\mathcal{L} = \sum_{i=1}^N \left[ \mathbb{E}_{q_\phi(\mathbf{E}_{\mathbf{x}_i}|\mathbf{x}_i)} \log p_{\boldsymbol{\theta}}(y_i|\mathbf{x}_i, \mathbf{E}_{\mathbf{x}_i}) - \mathrm{KL}(q_\phi(\mathbf{E}_{\mathbf{x}_i}|\mathbf{x}_i) \,\|\, p(\mathbf{E}_{\mathbf{x}_i})) \right]. \tag{6}$$

Here, $\boldsymbol{\theta}$ denotes the parameters of all $\mathbf{B}$ and $\mathbf{A}$ adapters across layers. Hence, $\phi = \{\boldsymbol{\theta}, \boldsymbol{\varphi}\}$ denotes all model parameters. Unlike standard variational inference as in (3), our formulation makes the distribution of $\mathbf{E}_{\mathbf{x}_i}$ input-dependent and replaces the single KL term with a sum of $N$ per-sample KL divergences, whose combined impact scales with $N$. With the objective defined in (6), we adopt a simple and fixed Gaussian prior $p(\mathbf{E}_{\mathbf{x}})$ for learning $\mathbf{E}_{\mathbf{x}}$ in each layer. The complete learning objective can be equivalently expressed as a summation over all samples: $\mathcal{L} = \sum_{(\mathbf{x},y)\in\mathcal{D}} \mathcal{L}(\mathbf{x}, y)$. Then, we learn the deterministic parameters $\mathbf{B}$ and $\mathbf{A}$ together represented by $\boldsymbol{\theta}$ using the expected negative log-likelihood (NLL) term while excluding the KL term. This ensures that learning of $\boldsymbol{\theta}$ remains supervised, yielding less noisy gradients per sample computed as

$$\nabla_{\boldsymbol{\theta}} \mathcal{L}(\mathbf{x}, y) = \mathbb{E}_{q_\phi(\mathbf{E}_{\mathbf{x}}|\mathbf{x})} \nabla_{\boldsymbol{\theta}} \log p_{\boldsymbol{\theta}}(y|\mathbf{x}, \mathbf{E}_{\mathbf{x}}). \tag{7}$$

We approximate this expectation with a single Monte Carlo draw $\mathbf{E}_{\mathbf{x}} \sim q_\phi(\mathbf{E}_{\mathbf{x}}|\mathbf{x})$ for each $\mathbf{x}$ sample.

Next, to update the auxiliary network parameters $\boldsymbol{\varphi}$, which appear in both the NLL and KL terms of (6), we apply the reparameterization trick [52] to handle random sampling of $\mathbf{E}_{\mathbf{x}}$. Concretely, in

each layer, $\mathbf{E}_{\mathbf{x}}^l \sim \mathcal{N}(\boldsymbol{\mu}_{\mathbf{E}}^l, \boldsymbol{\Omega}_{\mathbf{E}}^{l\,2})$ is reparameterized as $\mathbf{E}_{\mathbf{x}}^l = \boldsymbol{\mu}_{\mathbf{E}}^l + \boldsymbol{\Omega}_{\mathbf{E}}^l \odot \mathcal{E}^l$, where $\mathcal{E}^l \in \mathbb{R}^{r \times r}$ is sampled from $\mathcal{N}(\mathbf{0}, \mathbf{I})$. Thus, sampling $\mathbf{E}_{\mathbf{x}}^l$ from $q_\phi(\mathbf{E}_{\mathbf{x}}|\mathbf{x})$ is equivalent to evaluating a deterministic mapping $g_\phi(\mathcal{E}, \mathbf{x})$, enabling gradient computation via

$$\nabla_{\boldsymbol{\varphi}} \mathcal{L}(\mathbf{x}, y) = \mathbb{E}_{\mathcal{E} \sim \mathcal{N}(\mathbf{0}, \mathbf{I})} \left[ \nabla_{\boldsymbol{\varphi}} \left( \log p_{\boldsymbol{\theta}}(y|\mathbf{x}, g_\phi(\mathcal{E}, \mathbf{x})) - \log \frac{q_\phi(g_\phi(\mathcal{E}, \mathbf{x})|\mathbf{x})}{p(g_\phi(\mathcal{E}, \mathbf{x}))} \right) \right]. \qquad (8)$$

### 3.4 Posterior Predictive Inference and Model Complexity

**Posterior predictive inference and uncertainty estimation.** Once C-LoRA training has converged, we obtain point estimate of the fine-tuned model by replacing $\mathbf{E}_{\mathbf{x}}$ with its variational mean and computing $p(y|\mathbf{x}, \boldsymbol{\mu}_{\mathbf{E}}(\mathbf{x}))$. As shown in Section 4, we denote these point estimate models by setting $M$ in our model names. Next, to demonstrate the quality of uncertainty estimation, we sample $\mathbf{E}_{\mathbf{x}}$ from its inferred distribution $M$ times and, similar to (2), approximate the posterior predictive distribution using Monte Carlo sampling, yielding $p(y_{\text{test}}|\mathbf{x}_{\text{test}}, \mathcal{D}) = 1/M \sum_{m=1}^{M} p(y|\mathbf{x}, \mathbf{E}_{\mathbf{x}}^m)$, where $\mathbf{E}_{\mathbf{x}}^m \sim q_\phi(\mathbf{E}_{\mathbf{x}}|\mathbf{x})$. For example, for these calibrated models with $M = 10$ drawn samples in Section 4, we label them with "M=10" in our model names. When "M=0", the posterior mean is used directly for evaluation.

**Reducing contextual module complexity through feature reuse.** In layer $l$, the contextual module is designed to model $q_\phi(\mathbf{E}_{\mathbf{x}}^l|\mathbf{x}^{l-1})$ where $\mathbf{x}^l$ is derived from natural language; hence, learning features from scratch becomes both non-trivial and computationally expensive. To mitigate this issue, we follow [39] and take advantage of the main model by feeding $\mathbf{z}^l$ into the contextual module instead. Additional computational cost of $\mathcal{O}(r^4)$, stems from the auxiliary network $h_\varphi^l$, whose fully connected layers have size $C = \mathcal{O}(r^2) \ll d$. This overhead is minimal compared to the main model's per-layer operation cost of $\mathcal{O}(d^2)$. As a result, we can efficiently estimate uncertainty at the sample level in a low-dimensional space, sidestepping the costly feature-learning burden within the contextual modules during fine-tuning.

## 4 Experiments

In this section, we conduct comprehensive experiments to demonstrate the effectiveness of our proposed C-LoRA approach for uncertainty quantification in LLMs on reasoning datasets from various domains. We first specify the experimental settings in Section 4.1, including baselines, fine-tuning setting, and evaluation metrics. We then benchmark C-LoRA against existing uncertainty quantification methods across six reasoning datasets to evaluate overall task performance and uncertainty quality in Section 4.2. Additionally, we test the robustness of each method under distribution shift by fine-tuning on OBQA and evaluating on related OOD datasets in Section 4.3. Lastly, we perform an ablation to demonstrate the role of our auxiliary contextual module in Section 4.4.

### 4.1 Experimental Settings

**Baselines.** We provide performance comparisons of **C-LoRA** with well-established cutting-edge baselines that include Deep Ensemble [49, 51, 53], Monte-Carlo dropout (**MCD**) [29], Bayesian LoRA with Backpropagation (**BLoB**) [20], a recent mean-field VI approach applied to LoRA parameters; Laplace-LoRA (**LA**) [19], which uses a Laplace approximation over adapter weights; and *Maximum A Posteriori* (**MAP**), representing a deterministic point-estimate baseline.

**LLM fine-tuning settings.** Following prior work, we evaluate the performance of **C-LoRA**[1], by fine-tuning **LLaMA2-7B**[6] using the PEFT library[54] across six commonsense reasoning benchmarks, including both binary and multiple-choice classification tasks. For each task, the appropriate next-token logits are selected based on the format of the target output, which we include in the Appendix A, and the model is trained to maximize the log-likelihood of the correct answer. Following previous works [20, 21, 28], we apply LoRA to the query, value, and output projection layers using default hyperparameters. All models are fine-tuned with a batch size of 4. Baseline methods are trained for 5000 iterations, while **C-LoRA** is trained for 1500 iterations on all datasets, except for Winogrande-medium (WG-M), which is trained for 2000 iterations. To ensure consistent evaluation, we curate a training-validation split by reserving 80% of the original training set for fine-tuning and holding

---

[1]Using contextual modules comprising of two fully connected layers with 64 hidden units (C=64) each.

out the remaining 20% as a validation set. This validation split is used for early stopping and checkpointing based on a metric combining validation accuracy and calibration quality (see Appendix F).

**Evaluation metrics.** We report results using three key evaluation metrics: accuracy (ACC), expected calibration error (ECE), and negative log-likelihood (NLL). While ACC reflects raw predictive performance, ECE and NLL are widely adopted metrics for assessing the quality of uncertainty estimates. ECE measures the alignment between predicted confidence and actual correctness, while NLL evaluates the sharpness and correctness of the model's predicted probability distribution. All reported results are averaged over three independent runs with different random seeds, and we report the mean ± standard deviation for each metric.

## 4.2 Evaluation Results under In-distribution Fine-tuning

In this section, we conduct our assessment of fine-tuning performances under the in-distribution scenario for six common-sense reasoning tasks. These tasks encompass: Winogrande-small (`WG-S`), Winogrande-medium (`WG-M`) [55], ARC-Challenge (`ARC-C`), ARC-Easy (`ARC-E`) [56], Open-BookQA (`OBQA`)[57], and `BoolQ` [58]. For all the experiments, the same pre-trained LLaMA2-7B was used as the LLM backbone. For BLoB, we kept the same setting as in [20].

**Accuracy and Uncertainty Quantification**    Table 1 presents the performance of various uncertainty quantification methods applied to LoRA-tuned LLaMA2-7B across six common-sense reasoning datasets. As a complement, Figure 2 in Appendix K visualizes the results with bar plots. In this set of experiments, our proposed method, C-LoRA, as well as the baseline BLoB, are evaluated under both deterministic (M=0) and stochastic (M=10) posterior sampling settings. While C-LoRA, does not achieve the highest accuracy, it maintains competitive performance across all datasets, with accuracies that are typically within 1–2% of the best methods (e.g., `ARC-C`: 67.79%, `OBQA`: 78.26%). Notably, C-LoRA (M=0)—which uses the posterior mean without sampling—performs similarly or slightly better than the sampled version on some tasks (e.g., `ARC-C`: 69.02% vs. 67.79%), suggesting that the contextual posterior mean alone offers a robust deterministic approximation.

The strength of C-LoRA becomes more apparent when examining uncertainty calibration, as measured by expected calibration error (ECE). Here, C-LoRA (M=10) consistently achieves the lowest or second-lowest ECE across all datasets, including `WG-S` (6.86%), `ARC-C` (8.83%), `ARC-E` (4.27%), `WG-M` (3.71%), and `BoolQ` (1.62%). These values are significantly better than those obtained by MAP, MCD, and even Deep Ensemble, indicating that C-LoRA produces confidence estimates that are closely aligned with actual correctness. Even the deterministic variant, C-LoRA (M=0), outperforms BLoB (M=10) in calibration on datasets such as `WG-S` (7.16% vs. 11.23%) and performs comparably on `BoolQ` (1.46% vs. 1.46%). A similar pattern holds for negative log-likelihood (NLL), which reflects the quality of probabilistic predictions. C-LoRA (M=10) achieves among the lowest NLL values in nearly all cases, such as 0.63 on `WG-S`, and 0.88 on `ARC-C`, often outperforming both BLoB (M=10) and Deep Ensemble. Additionally, C-LoRA (M=0) matches or slightly outperforms BLoB (M=10) in NLL on several tasks (e.g., `ARC-C`: 0.89 vs. 0.88), while avoiding the computational overhead of posterior sampling. In summary, while C-LoRA trades off a small amount of accuracy compared to non-Bayesian methods, it delivers substantial improvements in calibration and likelihood, which are essential for uncertainty-aware applications.

**Post-hoc Calibration with Temperature Scaling**    To further evaluate calibration, we apply *post-hoc* temperature scaling [59] to both BLoB and C-LoRA and report the resulting ECE on six common-sense reasoning tasks in Table 2. Bar plots of these results are presented in Figure 3 in Appendix K. Even before temperature scaling, C-LoRA (M=10) achieves stronger calibration than BLoB on nearly every dataset. After applying temperature scaling, C-LoRA (M=0, tmp) achieves the best or second-best ECE in 4 of the 6 datasets, including `WG-S` (4.00%), `ARC-E` (3.86%), `ARC-C` (6.30%), and `WG-M` (2.90%). Notably, it outperforms BLoB (M=10, tmp) in 5 out of the 6 datasets, despite using no posterior sampling. Interestingly, C-LoRA (M=10, tmp) does not always show improvement over its uncalibrated variant. On datasets like `OBQA` and `BoolQ`, temperature scaling can actually worsen ECE (e.g., from 4.00% to 6.27% on `OBQA`). This may be due to the fact that posterior sampling already flattens the predictive distribution, and further scaling can overcorrect, leading to underconfident predictions. Also, temperature scaling minimizes NLL, a proper scoring rule, but it does not directly optimize ECE, which is a binned, non-differentiable, and improper

metric. Consequently, if a model is already reasonably well-calibrated or if its logit distribution exhibits certain local structures, temperature scaling may shift predictions in ways that degrade ECE in specific bins. Similar observations have been reported in prior work [60], where temperature scaling was found to degrade classwise ECE. These results suggest that C-LoRA (M=0) provides an effective balance between calibration quality and inference efficiency.

Table 1: Performance comparison of different methods applied to LoRA on LLAMA2-7B weights accross six common sense reasoning tasks with a validation set split from the training dataset. The best and the second best performances are specified in **boldface** and via underline respectively.

| Metric | Method | Datasets | | | | | |
|---|---|---|---|---|---|---|---|
| | | WG-S | ARC-C | ARC-E | WG-M | OBQA | BoolQ |
| ACC ↑ | MAP | $69.37_{\pm1.04}$ | $67.67_{\pm1.18}$ | $85.20_{\pm0.63}$ | $74.57_{\pm0.73}$ | $81.60_{\pm0.40}$ | $87.68_{\pm0.02}$ |
| | MCD | $69.06_{\pm1.40}$ | $66.66_{\pm2.30}$ | $85.49_{\pm0.74}$ | $75.89_{\pm0.48}$ | $81.46_{\pm0.92}$ | $87.67_{\pm0.08}$ |
| | Deep Ensemble | $68.98_{\pm0.97}$ | $68.57_{\pm2.11}$ | $86.24_{\pm1.26}$ | $77.39_{\pm1.08}$ | $82.20_{\pm0.91}$ | $88.07_{\pm0.17}$ |
| | LA | $68.18_{\pm1.04}$ | $64.17_{\pm0.97}$ | $85.30_{\pm0.97}$ | $74.15_{\pm0.40}$ | $77.53_{\pm0.80}$ | $86.45_{\pm0.35}$ |
| | BLoB (M=0) | $69.95_{\pm0.95}$ | $69.25_{\pm0.33}$ | $85.79_{\pm0.83}$ | $75.52_{\pm0.36}$ | $81.85_{\pm0.53}$ | $86.63_{\pm0.52}$ |
| | BLoB (M=10) | $66.55_{\pm0.61}$ | $66.66_{\pm2.25}$ | $84.56_{\pm0.20}$ | $73.38_{\pm0.29}$ | $81.44_{\pm0.53}$ | $86.63_{\pm0.50}$ |
| | C-LoRA (M=0) | $67.16_{\pm0.27}$ | $69.02_{\pm1.03}$ | $84.74_{\pm0.20}$ | $72.09_{\pm2.70}$ | $81.13_{\pm1.00}$ | $85.84_{\pm0.67}$ |
| | C-LoRA (M=10) | $66.21_{\pm1.24}$ | $67.79_{\pm1.27}$ | $84.38_{\pm0.67}$ | $70.48_{\pm1.71}$ | $78.26_{\pm2.61}$ | $84.64_{\pm0.81}$ |
| ECE ↓ | MAP | $29.76_{\pm1.08}$ | $30.60_{\pm1.26}$ | $13.49_{\pm0.63}$ | $23.01_{\pm0.44}$ | $15.30_{\pm0.11}$ | $5.93_{\pm0.36}$ |
| | MCD | $28.49_{\pm1.60}$ | $29.60_{\pm2.77}$ | $12.69_{\pm0.60}$ | $20.73_{\pm0.38}$ | $14.34_{\pm1.11}$ | $5.13_{\pm0.25}$ |
| | Deep Ensemble | $28.72_{\pm1.46}$ | $27.75_{\pm1.86}$ | $11.87_{\pm0.16}$ | $18.67_{\pm0.29}$ | $13.98_{\pm1.12}$ | $5.24_{\pm0.27}$ |
| | LA | $11.41_{\pm0.17}$ | $30.54_{\pm0.70}$ | $45.85_{\pm2.08}$ | $10.80_{\pm0.38}$ | $35.65_{\pm1.14}$ | $18.22_{\pm0.41}$ |
| | BLoB (M=0) | $21.22_{\pm1.67}$ | $22.57_{\pm1.24}$ | $10.13_{\pm0.39}$ | $12.35_{\pm0.86}$ | $9.35_{\pm1.08}$ | $2.90_{\pm0.18}$ |
| | BLoB (M=10) | $11.23_{\pm1.45}$ | $10.77_{\pm1.91}$ | $4.29_{\pm1.08}$ | $4.52_{\pm0.91}$ | **$3.82_{\pm0.96}$** | **$1.46_{\pm0.36}$** |
| | C-LoRA (M=0) | $7.16_{\pm2.92}$ | $12.28_{\pm1.01}$ | $5.75_{\pm1.00}$ | $6.07_{\pm4.85}$ | $5.10_{\pm1.36}$ | **$1.46_{\pm0.85}$** |
| | C-LoRA (M=10) | **$6.86_{\pm3.99}$** | **$8.83_{\pm1.20}$** | **$4.27_{\pm1.24}$** | **$3.71_{\pm1.30}$** | $4.00_{\pm0.84}$ | $1.62_{\pm0.44}$ |
| NLL ↓ | MAP | $2.86_{\pm0.23}$ | $3.07_{\pm0.09}$ | $1.13_{\pm0.10}$ | $1.26_{\pm0.12}$ | $1.04_{\pm0.02}$ | $0.34_{\pm0.00}$ |
| | MCD | $2.50_{\pm0.12}$ | $2.81_{\pm0.25}$ | $1.13_{\pm0.04}$ | $1.16_{\pm0.03}$ | $1.01_{\pm0.07}$ | $0.32_{\pm0.00}$ |
| | Deep Ensemble | $2.44_{\pm0.23}$ | $2.20_{\pm0.03}$ | $0.91_{\pm0.05}$ | $1.04_{\pm0.09}$ | $0.87_{\pm0.03}$ | $0.32_{\pm0.00}$ |
| | LA | **$0.62_{\pm0.00}$** | $1.17_{\pm0.01}$ | $0.97_{\pm0.05}$ | $0.56_{\pm0.00}$ | $0.98_{\pm0.01}$ | $0.45_{\pm0.00}$ |
| | BLoB (M=0) | $0.90_{\pm0.07}$ | $1.34_{\pm0.05}$ | $0.63_{\pm0.02}$ | $0.61_{\pm0.03}$ | $0.59_{\pm0.02}$ | **$0.31_{\pm0.00}$** |
| | BLoB (M=10) | $0.66_{\pm0.01}$ | **$0.88_{\pm0.03}$** | **$0.44_{\pm0.00}$** | **$0.54_{\pm0.00}$** | **$0.51_{\pm0.01}$** | **$0.31_{\pm0.01}$** |
| | C-LoRA (M=0) | $0.64_{\pm0.03}$ | $0.89_{\pm0.09}$ | $0.46_{\pm0.01}$ | $0.58_{\pm0.03}$ | $0.53_{\pm0.01}$ | $0.34_{\pm0.01}$ |
| | C-LoRA (M=10) | $0.63_{\pm0.02}$ | **$0.88_{\pm0.00}$** | $0.48_{\pm0.02}$ | $0.57_{\pm0.03}$ | $0.59_{\pm0.05}$ | $0.35_{\pm0.02}$ |

Table 2: Performance comparison of uncertainty quantification based on the ECE metric with temperature scaling using validation sets over six common-sense reasoning tasks.

| Metric | Method | Datasets | | | | | |
|---|---|---|---|---|---|---|---|
| | | WG-S | ARC-C | ARC-E | WG-M | OBQA | BoolQ |
| ECE ↓ | BLoB (M=0) | $21.22_{\pm1.67}$ | $22.57_{\pm1.24}$ | $10.13_{\pm0.39}$ | $12.35_{\pm0.86}$ | $9.35_{\pm1.08}$ | $2.90_{\pm0.18}$ |
| | BLoB (M=0, tmp) | $13.54_{\pm1.19}$ | $15.47_{\pm0.73}$ | $6.16_{\pm0.40}$ | $5.76_{\pm0.40}$ | $4.56_{\pm0.95}$ | $1.89_{\pm0.37}$ |
| | BLoB (M=10) | $11.23_{\pm1.45}$ | $10.77_{\pm1.91}$ | $4.29_{\pm1.08}$ | $4.52_{\pm0.91}$ | **$3.82_{\pm0.96}$** | **$1.46_{\pm0.36}$** |
| | BLoB (M=10, tmp) | $5.10_{\pm1.47}$ | $6.87_{\pm1.91}$ | $5.38_{\pm1.08}$ | $3.18_{\pm0.62}$ | $5.71_{\pm0.80}$ | $3.40_{\pm0.67}$ |
| | C-LoRA (M=0) | $7.16_{\pm2.92}$ | $12.28_{\pm1.01}$ | $5.75_{\pm1.00}$ | $6.07_{\pm4.85}$ | $5.10_{\pm1.36}$ | **$1.46_{\pm0.85}$** |
| | C-LoRA (M=0, tmp) | **$4.00_{\pm1.37}$** | $6.58_{\pm0.76}$ | **$3.86_{\pm0.17}$** | **$2.90_{\pm0.12}$** | $4.90_{\pm2.16}$ | $1.78_{\pm0.56}$ |
| | C-LoRA (M=10) | $6.86_{\pm3.99}$ | $8.83_{\pm1.20}$ | $4.27_{\pm1.24}$ | $3.71_{\pm1.30}$ | $4.00_{\pm0.84}$ | $1.62_{\pm0.44}$ |
| | C-LoRA (M=10, tmp) | $5.18_{\pm1.54}$ | **$6.30_{\pm0.87}$** | $4.35_{\pm1.35}$ | $3.87_{\pm1.77}$ | $6.27_{\pm1.01}$ | $2.67_{\pm0.29}$ |

## 4.3 Robustness Under Distribution Shift

Table 3 presents model performance under both in-distribution (OBQA) and out-of-distribution (OOD) settings with smaller (ARC-E, ARC-C) and larger (Chem, Phy) distribution shifts. Figure 4 in Appendix K illustrates bar plots of the results. While C-LoRA does not achieve the highest accuracy under shift, it demonstrates strong robustness in uncertainty estimation, especially in calibration (ECE) and likelihood (NLL). In terms of accuracy, Deep Ensemble and BLoB generally maintain higher predictive performance across OOD datasets. However, C-LoRA (M=10) remains competitive, achieving 65.09% accuracy on ARC-C and 41.00% on Chem—nearly matching the highest-performing methods. While C-LoRA (M=10) sees a slight drop in accuracy under severe shift (e.g., Phy: 31.00%),

Table 3: Performance comparison on out-of-distribution datasets. The following results are evaluated using LLaMA2-7B fine-tuned on the OBQA dataset in Table 1.

| Metric | Method | Datasets | | | | |
| | | *In-Dist.* | *Smaller Dist. Shift* | | *Larger Dist. Shift* | |
| | | OBQA | ARC-C | ARC-E | Chem | Phy |
| ACC ↑ | MAP | $81.60\pm_{0.40}$ | $67.67\pm_{1.52}$ | $73.88\pm_{0.27}$ | $41.00\pm_{3.60}$ | $33.00\pm_{5.29}$ |
| | MCD | $81.46\pm_{0.92}$ | $68.69\pm_{0.85}$ | $75.00\pm_{2.13}$ | $39.00\pm_{3.00}$ | $31.00\pm_{6.24}$ |
| | Deep Ensemble | $82.20\pm_{0.91}$ | $68.01\pm_{1.28}$ | $74.05\pm_{0.50}$ | $42.33\pm_{3.51}$ | $27.66\pm_{2.51}$ |
| | BLoB (M=0) | $81.85\pm_{0.53}$ | $69.48\pm_{0.78}$ | $76.93\pm_{1.33}$ | $45.33\pm_{1.15}$ | $26.66\pm_{4.61}$ |
| | BLoB (M=10) | $81.44\pm_{0.53}$ | $67.22\pm_{1.17}$ | $75.11\pm_{1.25}$ | $44.00\pm_{0.00}$ | $33.66\pm_{2.08}$ |
| | C-LoRA (M=0) | $81.13\pm_{1.00}$ | $66.66\pm_{0.78}$ | $75.99\pm_{1.63}$ | $40.00\pm_{1.73}$ | $28.00\pm_{1.73}$ |
| | C-LoRA (M=10) | $78.26\pm_{2.61}$ | $65.09\pm_{4.68}$ | $74.29\pm_{2.90}$ | $41.00\pm_{2.64}$ | $31.00\pm_{1.00}$ |
| ECE ↓ | MAP | $15.30\pm_{0.11}$ | $25.54\pm_{1.25}$ | $20.09\pm_{0.53}$ | $29.73\pm_{0.30}$ | $36.22\pm_{3.60}$ |
| | MCD | $14.34\pm_{1.11}$ | $23.41\pm_{0.74}$ | $18.36\pm_{1.03}$ | $28.67\pm_{0.77}$ | $36.53\pm_{4.30}$ |
| | Deep Ensemble | $13.98\pm_{1.12}$ | $20.90\pm_{1.05}$ | $16.89\pm_{0.80}$ | $16.10\pm_{2.22}$ | $26.74\pm_{3.23}$ |
| | BLoB (M=0) | $9.35\pm_{1.08}$ | $15.76\pm_{1.42}$ | $11.70\pm_{0.62}$ | $14.12\pm_{4.05}$ | $27.35\pm_{4.01}$ |
| | BLoB (M=10) | $3.82\pm_{0.96}$ | $\underline{9.77}\pm_{0.91}$ | $\mathbf{5.74}\pm_{0.91}$ | $\underline{12.63}\pm_{0.11}$ | $\mathbf{17.56}\pm_{2.81}$ |
| | C-LoRA (M=0) | $5.10\pm_{1.36}$ | $10.08\pm_{3.30}$ | $\underline{6.42}\pm_{2.48}$ | $15.42\pm_{2.99}$ | $24.42\pm_{6.02}$ |
| | C-LoRA (M=10) | $4.00\pm_{0.84}$ | $\mathbf{8.83}\pm_{1.44}$ | $6.68\pm_{0.71}$ | $\mathbf{12.49}\pm_{1.18}$ | $\underline{18.16}\pm_{1.52}$ |
| NLL ↓ | MAP | $1.04\pm_{0.02}$ | $1.66\pm_{0.12}$ | $1.37\pm_{0.04}$ | $1.81\pm_{0.03}$ | $1.86\pm_{0.04}$ |
| | MCD | $1.01\pm_{0.07}$ | $1.58\pm_{0.01}$ | $1.24\pm_{0.03}$ | $1.81\pm_{0.03}$ | $1.88\pm_{0.10}$ |
| | Deep Ensemble | $0.87\pm_{0.03}$ | $1.15\pm_{0.09}$ | $0.94\pm_{0.06}$ | $1.45\pm_{0.01}$ | $1.58\pm_{0.09}$ |
| | BLoB (M=0) | $0.59\pm_{0.02}$ | $0.95\pm_{0.05}$ | $0.72\pm_{0.01}$ | $1.41\pm_{0.02}$ | $1.57\pm_{0.03}$ |
| | BLoB (M=10) | $0.51\pm_{0.01}$ | $\mathbf{0.83}\pm_{0.02}$ | $\mathbf{0.63}\pm_{0.01}$ | $\underline{1.35}\pm_{0.01}$ | $\underline{1.46}\pm_{0.01}$ |
| | C-LoRA (M=0) | $0.53\pm_{0.01}$ | $\underline{0.85}\pm_{0.04}$ | $\underline{0.64}\pm_{0.04}$ | $1.43\pm_{0.01}$ | $1.54\pm_{0.09}$ |
| | C-LoRA (M=10) | $0.59\pm_{0.05}$ | $0.89\pm_{0.09}$ | $0.67\pm_{0.02}$ | $\mathbf{1.31}\pm_{0.02}$ | $\mathbf{1.42}\pm_{0.01}$ |

it significantly outperforms all baselines in ECE and NLL, indicating more reliable and calibrated uncertainty estimates.

For calibration (ECE), C-LoRA (M=10) achieves the lowest error on `Chem` (12.49%) and `ARC-C` (8.83%). This trend also holds for `Phy`, where C-LoRA (M=10) yields 18.16% which is only slightly worse than BLoB and significantly better than all other baselines. This indicates that C-LoRA remains well-calibrated even as the input distribution diverges from training. C-LoRA also excels in negative log-likelihood (NLL), achieving the lowest or second-lowest values across nearly all OOD datasets. On `Chem` and `Phy`, for example, C-LoRA (M=10) obtains NLL of 1.31 and 1.42 respectively, both lower than Deep Ensemble (1.45 and 1.58) and BLoB (1.35 and 1.46) and significantly better than all other baselines. Moreover, it is noteworthy that C-LoRA (M=0), which uses only the posterior mean, also maintains competitive performance. For example, on `ARC-C` and `ARC-E`, C-LoRA (M=0) achieves 10.08% and 6.42% ECE respectively, compared to BLoB (M=10)'s 9.77% and 5.74%. This suggests that C-LoRA's contextual posterior mean already captures meaningful uncertainty, making it attractive for deployment in compute-constrained scenarios.

## 4.4 Ablation Study: Impact of Contextual Module

We also conduct the ablation study investigating the importance of the contextual module for overall performance. Since in C-LoRA we focus on contextualizing the distribution of matrix $\mathbf{E} \in \mathbb{R}^{r \times r}$, to examine the effect of the contextual module, we compare with the results of fine-tuning the model on different tasks using lightweight factorization introduced in Section 3.1, with mean-field variational inference without the contextual module; we refer to this setting as **FE** in Tables 4 and 5.

Table 4 underscores the importance of the contextual module by comparing FE and C-LoRA across the six common-sense reasoning tasks under in-distribution scenario. C-LoRA offers significant improvements with respect to ECE across almost all tasks for inference-time sample with both M=0 and M=10, except for `BoolQ` where it achieves the second best performance. Also, C-LoRA gives the lowest NLL across all tasks.

Furthermore, we assess the effect of the contextual module on generalization. Table 5 provides the performance comparison of the corresponding models fine-tuned on `OBQA` in Table 4 under similar distribution shifts as in Table 3. C-LoRA outperforms FE across all datasets in both smaller and larger distribution shifts, with respect to ECE. In addition, C-LoRA offers the lowest NLL on all tasks in both

distribution shifts. With respect to accuracy, in all tasks C-LoRA achieves a comparable performance, except for `Chem` where it has a subpar performance; however, this shortcoming is a result of the tradeoff between a generalized model with a proper uncertainty quantification, and an overconfident model with a poor uncertainty estimation. In summary, these results indicate the significance of our auxiliary contextual module, particularly on enhancing the uncertainty quantification and generalization while maintaining competitive predictive performance.

Table 4: Impact of the contextual module on performance. The performance comparison is conducted across six common-sense reasoning tasks with a validation set split from the training dataset.

| Metric | Method | Datasets | | | | | |
|---|---|---|---|---|---|---|---|
| | | WG-S | ARC-C | ARC-E | WG-M | OBQA | BoolQ |
| ACC ↑ | FE (M=0) | $65.45_{\pm1.36}$ | $68.35_{\pm1.02}$ | $85.32_{\pm0.39}$ | $73.47_{\pm2.36}$ | $80.46_{\pm0.11}$ | $84.70_{\pm0.45}$ |
| | FE (M=10) | $65.06_{\pm1.33}$ | $68.20_{\pm2.40}$ | $85.08_{\pm0.36}$ | $72.43_{\pm1.28}$ | $80.93_{\pm0.50}$ | $84.65_{\pm0.52}$ |
| | C-LoRA (M=0) | $67.16_{\pm0.27}$ | $69.02_{\pm1.03}$ | $84.74_{\pm0.20}$ | $72.09_{\pm2.70}$ | $81.13_{\pm1.00}$ | $85.84_{\pm0.67}$ |
| | C-LoRA (M=10) | $66.21_{\pm1.24}$ | $67.79_{\pm1.27}$ | $84.38_{\pm0.67}$ | $70.48_{\pm1.71}$ | $78.26_{\pm2.61}$ | $84.64_{\pm0.81}$ |
| ECE ↓ | FE (M=0) | $23.76_{\pm1.64}$ | $27.26_{\pm0.90}$ | $12.23_{\pm0.25}$ | $14.90_{\pm1.98}$ | $11.23_{\pm0.35}$ | $2.57_{\pm0.13}$ |
| | FE (M=10) | $18.12_{\pm1.52}$ | $19.60_{\pm2.42}$ | $9.54_{\pm0.47}$ | $12.21_{\pm1.87}$ | $8.42_{\pm0.50}$ | $\mathbf{1.34}_{\pm0.22}$ |
| | C-LoRA (M=0) | $\underline{7.16}_{\pm2.92}$ | $\underline{12.28}_{\pm1.01}$ | $\underline{5.75}_{\pm1.00}$ | $\underline{6.07}_{\pm4.85}$ | $\underline{5.10}_{\pm1.36}$ | $\underline{1.46}_{\pm0.85}$ |
| | C-LoRA (M=10) | $\mathbf{6.86}_{\pm3.99}$ | $\mathbf{8.83}_{\pm1.20}$ | $\mathbf{4.27}_{\pm1.24}$ | $\mathbf{3.71}_{\pm1.30}$ | $\mathbf{4.00}_{\pm0.84}$ | $1.62_{\pm0.44}$ |
| NLL ↓ | FE (M=0) | $0.89_{\pm0.07}$ | $1.85_{\pm0.11}$ | $0.85_{\pm0.05}$ | $0.69_{\pm0.03}$ | $0.68_{\pm0.04}$ | $\mathbf{0.34}_{\pm0.00}$ |
| | FE (M=10) | $0.78_{\pm0.04}$ | $1.35_{\pm0.12}$ | $0.65_{\pm0.02}$ | $0.63_{\pm0.02}$ | $0.60_{\pm0.03}$ | $\mathbf{0.34}_{\pm0.00}$ |
| | C-LoRA (M=0) | $\underline{0.64}_{\pm0.03}$ | $\underline{0.89}_{\pm0.09}$ | $\mathbf{0.46}_{\pm0.01}$ | $\underline{0.58}_{\pm0.03}$ | $\mathbf{0.53}_{\pm0.01}$ | $0.34_{\pm0.01}$ |
| | C-LoRA (M=10) | $\mathbf{0.63}_{\pm0.02}$ | $\mathbf{0.88}_{\pm0.00}$ | $\underline{0.48}_{\pm0.02}$ | $\mathbf{0.57}_{\pm0.03}$ | $\underline{0.59}_{\pm0.05}$ | $0.35_{\pm0.02}$ |

Table 5: Impact of the contextual module on generalization. The performance comparison is conducted on out-of-distribution datasets using LLaMA2-7B fine-tuned on OBQA dataset in Table 4.

| Metric | Method | Datasets | | | | |
|---|---|---|---|---|---|---|
| | | In-Dist. | Smaller Dist. Shift | | Larger Dist. Shift | |
| | | OBQA | ARC-C | ARC-E | Chem | Phy |
| ACC ↑ | FE (M=0) | $80.46_{\pm0.11}$ | $68.91_{\pm1.22}$ | $74.72_{\pm0.63}$ | $45.00_{\pm1.73}$ | $31.66_{\pm3.51}$ |
| | FE (M=10) | $80.93_{\pm0.50}$ | $66.65_{\pm1.09}$ | $74.76_{\pm0.70}$ | $46.00_{\pm1.73}$ | $32.33_{\pm1.52}$ |
| | C-LoRA (M=0) | $81.13_{\pm1.00}$ | $66.66_{\pm0.78}$ | $75.99_{\pm1.63}$ | $40.00_{\pm1.73}$ | $28.00_{\pm1.73}$ |
| | C-LoRA (M=10) | $78.26_{\pm2.61}$ | $65.09_{\pm4.68}$ | $74.29_{\pm2.90}$ | $41.00_{\pm2.64}$ | $31.00_{\pm1.00}$ |
| ECE ↓ | FE (M=0) | $11.23_{\pm0.35}$ | $18.44_{\pm1.51}$ | $14.65_{\pm0.49}$ | $20.02_{\pm2.26}$ | $31.75_{\pm3.20}$ |
| | FE (M=10) | $8.42_{\pm0.50}$ | $15.78_{\pm1.11}$ | $11.69_{\pm0.68}$ | $17.61_{\pm1.25}$ | $26.02_{\pm1.32}$ |
| | C-LoRA (M=0) | $5.10_{\pm1.36}$ | $\underline{10.08}_{\pm3.30}$ | $6.42_{\pm2.48}$ | $\underline{15.42}_{\pm2.99}$ | $\underline{24.42}_{\pm6.02}$ |
| | C-LoRA (M=10) | $4.00_{\pm0.84}$ | $\mathbf{8.83}_{\pm1.44}$ | $\underline{6.68}_{\pm0.71}$ | $\mathbf{12.49}_{\pm1.18}$ | $\mathbf{18.16}_{\pm1.52}$ |
| NLL ↓ | FE (M=0) | $0.68_{\pm0.04}$ | $1.03_{\pm0.01}$ | $0.87_{\pm0.01}$ | $1.53_{\pm0.05}$ | $1.77_{\pm0.03}$ |
| | FE (M=10) | $0.60_{\pm0.03}$ | $0.95_{\pm0.02}$ | $0.78_{\pm0.01}$ | $1.46_{\pm0.03}$ | $1.67_{\pm0.02}$ |
| | C-LoRA (M=0) | $0.53_{\pm0.01}$ | $\mathbf{0.85}_{\pm0.04}$ | $\mathbf{0.64}_{\pm0.04}$ | $\underline{1.43}_{\pm0.01}$ | $\underline{1.54}_{\pm0.09}$ |
| | C-LoRA (M=10) | $0.59_{\pm0.05}$ | $\underline{0.89}_{\pm0.09}$ | $\underline{0.67}_{\pm0.02}$ | $\mathbf{1.31}_{\pm0.02}$ | $\mathbf{1.42}_{\pm0.01}$ |

## 5 Related Works

LLMs have demonstrated remarkable successes across a wide range of applications, including code generation, scientific reasoning, and open-domain question answering; however, they are also notorious for *hallucination*—a phenomenon in which the model generates fluent but factually incorrect or unsupported content that is not grounded in the input or external knowledge. To mitigate hallucination, a variety of strategies have been proposed, including Reinforcement Learning from Human Feedback (RLHF) to align model outputs with human preferences [61, 62], and Retrieval-Augmented Generation (RAG) to ground responses in external knowledge [63, 64]. Additionally, probabilistic methods such as BNNs [19, 20] and conformal prediction [65, 66] have been demonstrated to be able to quantify uncertainty and flag unreliable outputs. Recent work also shows the effectiveness of prompt-level interventions [67, 68] by carefully crafting instructions and uncertainty-aware prompts to reduce hallucinated content without altering the model architecture.

Our work connects to previous efforts in BNNs by introducing a Bayesian formulation over the LoRA layers; however, we extend this line of work by using a data-dependent lightweight low-rank

factorization, where the intermediate matrix $\mathbf{E}$ is modeled as random variables, enabling flexible and context-aware uncertainty estimation to mitigate overconfidence and hallucination in LLMs.

# 6 Conclusion and Discussion

In this work, we introduced C-LoRA, a novel uncertainty-aware parameter-efficient approach for fine-tuning LLMs in an end-to-end Bayesian framework. C-LoRA facilitates modeling the aleatoric uncertainty (data uncertainty) via an efficient and scalable contextual module, without compromising the potential for estimating the model uncertainty. Through extensive empirical studies, we demonstrate the superior performance of our method in achieving outstanding accuracy and strong uncertainty quantification capabilities, with consistent generalization across diverse data distributions. Our method underscores the significance of modeling the aleatoric uncertainty in low-data regimes that can lead to substantial gains in generalization and trustworthiness of LLMs.

While C-LoRA establishes a promising foundation for uncertainty-aware parameter-efficient fine-tuning, several avenues remain open for future exploration. One interesting direction is extending the framework to multi-modal or multi-task settings, where capturing cross-modal or task-dependent uncertainty could further enhance robustness and generalization. Another promising avenue involves integrating C-LoRA with active learning or adaptive data acquisition strategies, leveraging its uncertainty estimates to guide sample-efficient model updates. Moreover, exploring hierarchical or structured priors within the Bayesian formulation may offer richer representations of both epistemic and aleatoric uncertainty.

# 7 Limitations

In this study, we restricted our experiments to fine-tuning LLaMA2-7B models, which limits our understanding of how C-LoRA scales to larger architectures. Investigating the scaling behavior of C-LoRA on larger models remains an open challenge. Furthermore, evaluating uncertainty in language models remains challenging due to the lack of standardized benchmarks with ground-truth uncertainty labels. While our evaluation—based on held-out test sets and established metrics like ECE and NLL—provides meaningful insight into model calibration and predictive confidence, these metrics offer only an indirect view of uncertainty quality. Developing task-specific benchmarks or human-in-the-loop protocols for evaluating uncertainty in NLP would be a valuable direction. Moreover, while C-LoRA is motivated by data-dependent/aleatoric uncertainty, this work does not provide a theoretical guarantee that it can disentangle aleatoric and epistemic uncertainty, which we leave to future investigation.

We also note that, as shown throughout our experiments, C-LoRA overall provides a more calibrated predictions and better uncertainty estimation at the expense of marginally reduced predictive performance which can be preferable in many high-stakes applications. However, in other domains such as spam detection where accuracy is more critical, this trade-off may not be desirable. Hence we acknowledge that this trade-off is task dependent and should be considered before deploying C-LoRA (and in general any uncertainty-aware method).

Finally, our approach employs amortized variational inference to approximate the posterior distribution over the LoRA adapter parameters, providing a scalable and principled Bayesian treatment suitable for large language models. This variational approximation, based on a Gaussian family and a reverse KL objective, is inherently mode-seeking and may underestimate posterior uncertainty or fail to capture multi-modality in the true posterior. Investigating richer variational families and more comprehensive posterior predictive analyses represents an important direction for future work.

# 8 Acknowledgment

A.H.R., W.Z., Y.W., and X.N.Q. acknowledge the support from United States National Science Foundation (NSF) grants DMREF-2119103, SHF-2215573, and IIS-2212419. S.J., B.J.Y., N.M.U., and X.N.Q. acknowledge the support from the United States Department of Energy's Office of Science Biological and Environmental Research (BER) program under project B&R# KP1601017 and FWP#CC140. Many of the numerical experiments were conducted using advanced computing resources provided by Texas A&M High Performance Research Computing.

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

# A  Dataset Details

A brief summary of the prompt templates used for fine-tuning LLaMA2 on common-sense reasoning tasks is provided in Table 6. More details on the size of training datasets after applying an $80\%$ train-validation split, and the number of labels are gathered in Table 7.

Table 6: Prompt templates for fine-tuning on common-sense reasoning tasks

| task | prompt |
|---|---|
| Winogrande (WG-S/WG-M) | Select one of the choices that answers the following question: {question} Choices: A. {option1}. B. {option2}. Answer: |
| ARC (ARC-C/ARC-E), Openbook QA (OBQA), MMLU | Select one of the choices that answers the following question: {question} Choices: A. {choice1}. B. {choice2}, C. {choice3}. D. {choice4}. Answer: |
| BoolQ | Answer the question with only True or False: {question} Context: {passage} |

Table 7: Size of the training dataset after train-validation split and number of labels in each dataset

|  | WG-S | ARC-C | ARC-E | WG-M | OBQA | BoolQ |
|---|---|---|---|---|---|---|
| Training dataset size | 512 | 892 | 1.8K | 2.05k | 3.97k | 1.99k |
| Number of labels | 2 | 5 | 5 | 2 | 4 | 2 |

# B  Temperature Scaling

Temperature scaling is a commonly adopted method to convert the predicted probabilities to more calibrated values [59, 69]. To do so, it employs a positive constant scaler, $T$, called the temperature parameter, to *soften* the softmax values making the distribution less peaky. That is, given the logit vector $\mathbf{z}$, the new confidence can be expressed as,

$$\mathbf{q} = \sigma(\mathbf{z}/T), \tag{9}$$

where $\sigma$ is the softmax operator. It has been empirically shown in [59] that such temperature scaling leads to lower Expected Calibration Error (ECE) on classification tasks. Here, $T$ can be estimated via optimizing the Negative Log-Likelihood (NLL) on the validation set. Since $T$ does not change the maximum of the softmax function, the prediction is unaffected; therefore, it does not change the model's accuracy while enhancing model's calibration.

# C  Uncertainty Estimation Metrics

Uncertainty quantification is commonly assessed by NLL and ECE in the literature. The summation of the negative expected log probability of predicting the correct label is calculated for NLL. That is, for the model $P_\theta$, and a dataset of size $N$, NLL is computed as,

$$\mathrm{NLL} = \frac{1}{N} \sum_{i=1}^{N} -\log P_\theta(y_i), \tag{10}$$

where $y_i$ denotes the correct label. This metric promotes the model to assign a higher probability to the correct predictions. For an overconfident model in an incorrect prediction, the probability of correct answer decreases, which leads to an increase in NLL. ECE on the other hand, estimates how well a model is calibrated by assessing how close the model's confidence is to its accuracy. Specifically, by binning the predictions according to the confidence levels, this metric is calculated by the weighted average of the absolute difference in each bin, that is,

$$\mathrm{ECE} = \sum_{m=1}^{M} \frac{|B_m|}{N} |\mathrm{acc}(B_m) - \mathrm{conf}(B_m)|, \tag{11}$$

where acc($B_m$) and conf($B_m$) indicate the average accuracy and confidence in each bin, $B_m$,

$$\text{acc}(B_m) = \frac{1}{|B_m|} \sum_{i \in B_m} \mathbf{1}(\hat{y}_i = y_i), \quad \text{conf}(B_m) = \frac{1}{|B_m|} \sum_{i \in B_m} P(\hat{y}_i), \tag{12}$$

in which $|B_m|$ denotes the number of examples in bin $m$.

## D   Hyperparameters

Table 8 summarizes the hyperparameters for LoRA fine-tuning of LLaMA2-7B, selected based on prior work [19–21] and default values provided in PEFT library [54]. Following [20], we optimize the KL term with SGD and a linear learning-rate scheduler.

Table 8: Hyperparameters of LoRA fine-tuning of LlaMA2-7B

| Hyper-parameter | Value |
|---|---|
| Optimizer | AdamW |
| LR Scheduler | Linear |
| Learning Rate | $1e$-4 |
| Batch Size | 4 |
| Max Sequence Length | 300 |
| LoRA $\alpha$ | 16 |
| LoRA $r$ | 8 |

## E   On the Importance of Flexibility

To study the impact of complexity we examine the performance of FE with the case where the matrix **E** is a diagonal matrix with $r$ random variables. We refer to this as **DE**.

Table 9 summarizes the performance of DE and FE with M=0 and M=10 under in-distribution scenario for the six common-sense reasoning tasks. These models are ordered from top to bottom for each metric, such that each successive model is more flexible than the previous one. From Table 9, it is clear that as the flexibility increases, the performance generally improves. Specifically, in almost all tasks, FE provides a better uncertainty quantification with respect to DE, with a lower ECE and NLL. Although FE does not always deliver the highest accuracy, it shows a comparable performance across all tasks. Particularly, it provides the best accuracy in `ARC-C` under both deterministic (M=0) and stochastic (M=10) settings. It can be concluded from Table 9 that a more flexible model yields more reliable uncertainty quantification, with no significant loss in predictive accuracy.

To assess the effect of model complexity on generalization, we further compared DE and FE under out-of-distribution scenario, using the models fine-tuned on `OBQA` from Table 9. As Table 10 indicates, FE consistently delivers superior uncertainty quantification over all tasks under both smaller and larger distribution shifts. It also delivers the best accuracy in almost all tasks.

These findings from Tables 9 and 10 confirm that richer stochastic structures enhance both generalization and uncertainty estimation, motivating our focus on contextualizing the full matrix **E**.

## F   Checkpoint Metric

Given our goal of achieving a well-calibrated model while maintaining competitive accuracy relative to the states of the art (SOTAs), we devise a metric that incorporates both accuracy (ACC) and expected calibration error (ECE). Specifically, we define:

$$\mathcal{C} = (1 - \text{ACC}_{\text{val}}) \cdot \text{ECE}_{\text{val}},$$

with $\text{ACC}_{\text{val}}$ and $\text{ECE}_{\text{val}}$ representing the validation accuracy and expected calibration error, respectively. During training, the model is evaluated using this criterion every 100 steps. We summarize the training algorithm of C-LoRA in Algorithm 1.

Table 9: Impact of different levels of flexibility on performance. Performance comparison is conducted across six common-sense reasoning tasks with a validation set split from the training dataset. The best and the second best performances are specified in **boldface** and via underline respectively.

| Metric | Method | Datasets | | | | | |
|---|---|---|---|---|---|---|---|
| | | WG-S | ARC-C | ARC-E | WG-M | OBQA | BoolQ |
| ACC ↑ | DE (M=0) | $65.98\pm_{2.46}$ | $66.77\pm_{1.28}$ | $85.61\pm_{0.36}$ | $75.19\pm_{0.05}$ | $81.80\pm_{1.05}$ | $87.47\pm_{0.03}$ |
| | DE (M=10) | $66.09\pm_{1.50}$ | $66.88\pm_{1.47}$ | $85.71\pm_{0.64}$ | $75.19\pm_{0.16}$ | $82.06\pm_{0.41}$ | $87.44\pm_{0.07}$ |
| | FE (M=0) | $65.45\pm_{1.36}$ | $68.35\pm_{1.02}$ | $85.32\pm_{0.39}$ | $73.47\pm_{2.36}$ | $80.46\pm_{0.11}$ | $84.70\pm_{0.45}$ |
| | FE (M=10) | $65.06\pm_{1.33}$ | $68.20\pm_{2.40}$ | $85.08\pm_{0.36}$ | $72.43\pm_{1.28}$ | $80.93\pm_{0.50}$ | $84.65\pm_{0.52}$ |
| ECE ↓ | DE (M=0) | $30.99\pm_{1.39}$ | $29.63\pm_{1.58}$ | $13.17\pm_{0.30}$ | $21.91\pm_{0.49}$ | $14.19\pm_{0.37}$ | $4.67\pm_{0.17}$ |
| | DE (M=10) | $27.00\pm_{0.62}$ | $26.26\pm_{1.17}$ | $11.79\pm_{0.16}$ | $20.21\pm_{0.09}$ | $13.24\pm_{0.34}$ | $4.41\pm_{0.02}$ |
| | FE (M=0) | $23.76\pm_{1.64}$ | $27.26\pm_{0.90}$ | $12.23\pm_{0.25}$ | $14.90\pm_{1.98}$ | $11.23\pm_{0.35}$ | $2.57\pm_{0.13}$ |
| | FE (M=10) | $\mathbf{18.12\pm_{1.52}}$ | $\mathbf{19.60\pm_{2.42}}$ | $\mathbf{9.54\pm_{0.47}}$ | $\mathbf{12.21\pm_{1.87}}$ | $\mathbf{8.42\pm_{0.50}}$ | $\mathbf{1.34\pm_{0.22}}$ |
| NLL ↓ | DE (M=0) | $2.14\pm_{0.00}$ | $2.83\pm_{0.26}$ | $1.17\pm_{0.08}$ | $1.18\pm_{0.06}$ | $0.91\pm_{0.02}$ | $\mathbf{0.32\pm_{0.00}}$ |
| | DE (M=10) | $1.58\pm_{0.05}$ | $2.21\pm_{0.06}$ | $1.05\pm_{0.07}$ | $1.05\pm_{0.05}$ | $0.85\pm_{0.02}$ | $\mathbf{0.32\pm_{0.00}}$ |
| | FE (M=0) | $0.89\pm_{0.07}$ | $1.85\pm_{0.11}$ | $0.85\pm_{0.05}$ | $0.69\pm_{0.03}$ | $0.68\pm_{0.04}$ | $0.34\pm_{0.00}$ |
| | FE (M=10) | $\mathbf{0.78\pm_{0.04}}$ | $\mathbf{1.35\pm_{0.12}}$ | $\mathbf{0.65\pm_{0.02}}$ | $\mathbf{0.63\pm_{0.02}}$ | $\mathbf{0.60\pm_{0.03}}$ | $0.34\pm_{0.00}$ |

Table 10: Impact of different levels of complexity on generalization. Performance comparison is conducted on out-of-distribution datasets. The following results are evaluated using LLaMA2-7B fine-tuned on the OBQA dataset. The best and the second best performances are specified in **boldface** and via *underline* respectively.

| Metric | Method | Datasets | | | | |
|---|---|---|---|---|---|---|
| | | *In-Dist.* | *Smaller Dist. Shift* | | *Larger Dist. Shift* | |
| | | OBQA | ARC-C | ARC-E | Chem | Phy |
| ACC ↑ | DE (M=0) | $81.80\pm_{1.05}$ | $68.46\pm_{1.92}$ | $74.75\pm_{0.79}$ | $44.33\pm_{2.08}$ | $31.00\pm_{4.35}$ |
| | DE (M=10) | $82.06\pm_{0.41}$ | $67.90\pm_{0.67}$ | $74.80\pm_{1.07}$ | $43.33\pm_{0.57}$ | $30.66\pm_{4.51}$ |
| | FE (M=0) | $80.46\pm_{0.11}$ | $68.91\pm_{1.22}$ | $74.72\pm_{0.63}$ | $45.00\pm_{1.73}$ | $31.66\pm_{3.51}$ |
| | FE (M=10) | $80.93\pm_{0.50}$ | $66.65\pm_{1.09}$ | $74.76\pm_{0.70}$ | $46.00\pm_{1.73}$ | $32.33\pm_{1.52}$ |
| ECE ↓ | DE (M=0) | $14.19\pm_{0.37}$ | $23.51\pm_{2.07}$ | $17.74\pm_{0.64}$ | $23.37\pm_{3.97}$ | $34.33\pm_{2.34}$ |
| | DE (M=10) | $13.24\pm_{0.34}$ | $22.31\pm_{0.53}$ | $16.81\pm_{1.17}$ | $21.90\pm_{1.90}$ | $35.14\pm_{5.68}$ |
| | FE (M=0) | $11.23\pm_{0.35}$ | $18.44\pm_{1.51}$ | $14.65\pm_{0.49}$ | $20.02\pm_{2.26}$ | $31.75\pm_{3.20}$ |
| | FE (M=10) | $8.42\pm_{0.50}$ | $\mathbf{15.78\pm_{1.11}}$ | $\mathbf{11.69\pm_{0.68}}$ | $\mathbf{17.61\pm_{1.25}}$ | $\mathbf{26.02\pm_{1.32}}$ |
| NLL ↓ | DE (M=0) | $0.91\pm_{0.02}$ | $1.35\pm_{0.08}$ | $1.08\pm_{0.03}$ | $1.64\pm_{0.06}$ | $1.86\pm_{0.05}$ |
| | DE (M=10) | $0.85\pm_{0.02}$ | $1.28\pm_{0.05}$ | $1.04\pm_{0.02}$ | $1.62\pm_{0.06}$ | $1.82\pm_{0.05}$ |
| | FE (M=0) | $0.68\pm_{0.04}$ | $1.03\pm_{0.01}$ | $0.87\pm_{0.01}$ | $1.53\pm_{0.05}$ | $1.77\pm_{0.03}$ |
| | FE (M=10) | $0.60\pm_{0.03}$ | $\mathbf{0.95\pm_{0.02}}$ | $\mathbf{0.78\pm_{0.01}}$ | $\mathbf{1.46\pm_{0.03}}$ | $\mathbf{1.67\pm_{0.02}}$ |

## G Flipout

To speed up the sampling, we apply the Flipout technique–originally introduced in [70] and also adopted by [51]–in C-LoRA to the low-rank matrix $\mathbf{E}$. In particular, having two randomly sampled flipping vectors $s = \{-1, +1\}^r$ and $t = \{-1, +1\}^r$, and considering $\mathbf{b}_i$ to be the $i$-th input in a mini-batch, the output after flipout is:

$$\mathbf{o_i} = \mathbf{W}\mathbf{b}_i = \mathbf{W_0}\mathbf{b}_i + \mathbf{BEA}\mathbf{b}_i = \mathbf{W_0}\mathbf{b}_i + \mathbf{B}(\boldsymbol{\mu_E} + (\mathcal{E} \circ \boldsymbol{\Omega_E}) \circ (t_i s_i^\top))\mathbf{A}\mathbf{b}_i, \quad \mathcal{E} \sim \mathcal{N}(\mathbf{0}, \mathbf{I})$$

## H Discussion on Results of Laplace Approximation

For experiments involving fine-tuning via Laplace approximation (LA), we used the publicly released code provided by the authors of [19]. However, the results that we attempted to reproduce—reported in Table 1 in our main paper—are substantially worse than those reported in the original paper [19]. This discrepancy is especially pronounced for the OBQA and BoolQ datasets. Our findings are

---

**Algorithm 1** Contextual Low-rank Adaptation **(C-LoRA)**

---

**Require:** Dataset split: $\mathcal{D}_{\text{train}}, \mathcal{D}_{\text{validation}}, \mathcal{D}_{\text{test}}$
**Require:** contextual module $h_\varphi$, deterministic parameters $\boldsymbol{\theta}$, number of iterations $T$, evaluation frequency $f_{\text{eval}}$, learning rate $\eta$, checkpoint metric $b$
  1: $b \leftarrow \infty$
  2: **for** $t \leftarrow 0$ to $T$ **do**
  3:      $\mathbf{x}, y \sim \mathcal{D}_{\text{train}}$
  4:      $\mathbf{x}^0 \leftarrow \mathbf{x}$
  5:      **for** $l \leftarrow 1$ to $L$ **do**
  6:          $\boldsymbol{\mu}_{\mathbf{E}}^l, \boldsymbol{\Omega}_{\mathbf{E}}^l \leftarrow h_\varphi^l(\mathbf{x}^{l-1})$
  7:          $\mathcal{E}^l \sim \mathcal{N}(\mathbf{0}, \mathbf{I})$
  8:          $\mathbf{E}_{\mathbf{x}}^l \leftarrow \text{Flipout}(\mathcal{E}^l)$
  9:      **end for**
10:      $\boldsymbol{\theta} \leftarrow \boldsymbol{\theta} - \eta \nabla_{\boldsymbol{\theta}} \mathcal{L}(\mathbf{x}, y)$                                 ▷ Eq. 7
11:      $\varphi \leftarrow \varphi - \eta \nabla_{\varphi} \mathcal{L}(\mathbf{x}, y)$                                ▷ Eq. 8
12:      **if** $t \bmod f_{\text{eval}} = 0$ **then**
13:          Compute validation accuracy and ECE
14:          $\tilde{b} \leftarrow (1 - \text{ACC}_{\text{val}}) \cdot \text{ECE}_{\text{val}}$
15:          **if** $\tilde{b} < b$ **then**
16:              $b \leftarrow \tilde{b}$
17:              Save $\boldsymbol{\theta}, \varphi$                        ▷ Checkpoint best model
18:              Record performance on $\mathcal{D}_{\text{test}}$
19:          **end if**
20:      **end if**
21: **end for**

---

consistent with those reported by the authors of BLoB [20], which suggests that the degradation in performance may stem from sub-optimal MAP solutions that negatively impact the quality of the Laplace approximation-based LoRA fine-tuning.

Additionally, in our experiments, we observed that LA is notably memory-intensive, requiring hardware with higher memory capacity than BLoB and our C-LoRA implementations. This might make the practical applicability of LA more challenging considering the scalability compared to other competing methods.

# I   Discussion on the choice of prior

In this work, we adopt a fixed Gaussian prior which, when paired with a variational Gaussian posterior, enables a closed-form KL divergence and further facilitates performance comparison across baseline methods. However, due to Monte Carlo approximation of the ELBO, alternative richer distributions could also be considered as priors. While richer priors may offer empirical benefits in specific settings, they can introduce additional computational and design challenges. Moreover, due to the highly expressive contextual modules, despite the simplicity, fixed Gaussian prior mainly act as a light regularizer rather than as a strict inductive bottleneck. Prior work [71] has also shown that Bayesian Model Averaging (BMA) is robust to the choice of prior, with posterior predictive behavior remaining similar across different prior families.

# J   Empirical Study of Contextualized Variance

To further illustrate that our model captures *input-dependent* uncertainty, we analyze the predictive variance behavior of the LM head for three OBQA questions, each sharing the same correct label (option "C"). For each question, we extract the predicted variance matrix from the final layer, compute the mean variance across tokens, and report the $\ell_2$ norm of the resulting vector to summarize overall uncertainty.

Table 11: **Example questions and corresponding variance magnitudes.** Despite sharing the same label, the model assigns distinct variances to each question, reflecting input-dependent uncertainty.

| Question | Variance $\ell_2$ norm |
|---|---|
| Q1: What would light bounce off of when it hits it? | 3.7661 |
| Q2: A mother births what? | 3.6792 |
| Q3: What happens when animals in hot environments are active? | 3.9443 |

Despite sharing the same label, the model assigns distinct variances to each question, reflecting differences in ambiguity and specificity of the input. This confirms that C-LoRA modulates its predictive distribution based on input semantics, not merely on class labels.

We further examined whether the learned variance meaningfully affects the model's representations. Specifically, we computed the $\ell_2$ norm of the difference between the last-token embedding at $M = 0$ and the mean embedding across $M = 10$ stochastic samples. For the three representative examples (Q1–Q3), the differences ranged from 6.18 to 7.25, while the embedding norms themselves were approximately 120–122, corresponding to a relative change of about 5–6%. In embedding space, this level of perturbation is non-trivial and indicates meaningful input-dependent variability introduced by the contextualized variance modules.

We extended this analysis to a broader set of examples, comparing 8 correct and 8 incorrect predictions. For each, we computed the $\ell_2$ norm of the difference between the deterministic embedding ($M = 0$) and the mean of embeddings from $M = 10$ samples, then averaged across each group. Correct predictions had an average difference of $6.7 \pm 1.09$, while incorrect predictions showed higher variability at $8.45 \pm 2.66$. This systematic increase in embedding perturbation for incorrect predictions suggests that C-LoRA's learned variance corresponds to greater uncertainty for ambiguous or difficult inputs, further supporting its interpretation as modeling input-specific uncertainty.

## K   Visualization of Results

For a visual performance comparison, we present the results under in-distribution scenario, after temperature scaling, and the results under out-of-distribution scenario, reported in Tables 1, 2, and 3, using bar plots.

Figure 3 presents bar plots summarizing the results reported in Table 1. As discussed in Section 4.2, all methods exhibit similar predictive performance overall, while C-LoRA and BLoB ($M = 10$) achieve superior uncertainty quantification, as measured by ECE and NLL. Figure 3 also illustrates the effect of temperature scaling on calibration, corresponding to Table 2. As noted in Section 4.2, temperature scaling substantially improves calibration. Finally, Figure 4 visualizes the out-of-distribution results from Table 3 via bar plots. Consistent with in-distribution results, performance degrades as the distribution shifts; however, predictive performance remains largely similar across methods, with C-LoRA and BLoB demonstrating more robust uncertainty quantification according to ECE and NLL. As we mentioned earlier, it worth noting that, although C-LoRA offers similar predictive performance overall, it can underperform on certain datasets in exchange for better uncertainty quantification.

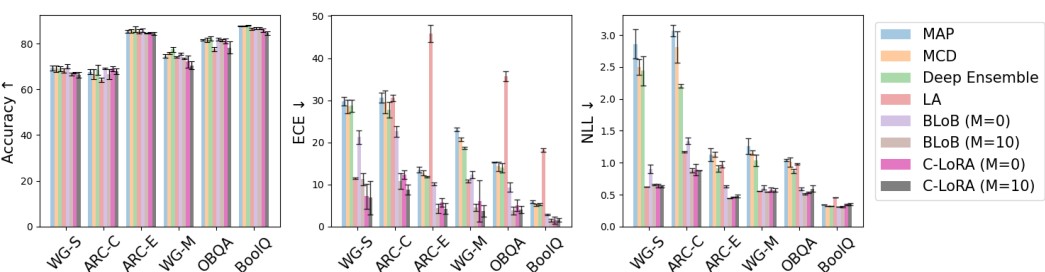

Figure 2: Visualization of the results in Table 1.

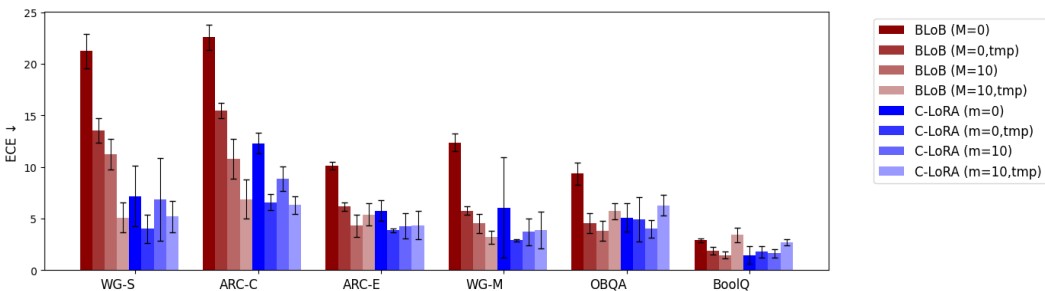

Figure 3: Visualization of the results in Table 2.

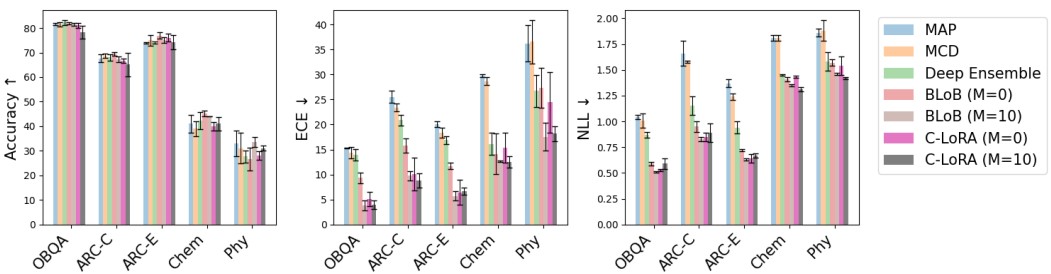

Figure 4: Visualization of the results in Table 3.

## L   Broader Impacts

Our uncertainty-aware fine-tuning framework integrating uncertainty quantification in LLMs could enhance the safety, reliability, and trustworthiness of AI systems deployed in high-stakes settings such as medical diagnosis, atmospheric modeling, and autonomous navigation, where unrecognized model errors can have major consequences. Moreover, as our approach explicitly models aleatoric or data uncertainty, it is particularly well-suited for low-resource tasks and rare-event prediction, providing access to robust LLM-based AI tools in fields ranging from global health to societal governance and policy-making.

We anticipate minimal additional computational overhead compared to standard fine-tuning, and because our framework is compatible with existing Bayesian extensions for epistemic uncertainty, it offers a clear path toward unified uncertainty quantification without introducing undue complexity. We do not foresee any negative societal impacts beyond those already inherent to large-scale model deployment, and we believe that equipping models with calibrated confidence measures is an essential step toward more ethical, accountable, and human-centered AI.

## M   Additional Information

In our contextual module, we set $C$, the number of hidden units, to $64$ in order to ensure sufficient expressiveness for learning meaningful features. Also, to have a robust performance, the variance output, $\Omega_{\mathbf{E}}$, uses a sigmoid activation function; however, we do not use any activation function for the mean $\mu_{\mathbf{E}}$. All the experiments for almost all methods were conducted using 1 NVIDIA A100 GPU with $40$ GB memory except for BLoB, for which we used 2 NVIDIA A100 with $40$ GB memory. Also for LA we used NVIDIA L40S GPU with $48$ GB memory due to its memory demands.

