# OpenReview forum: "C-LoRA: Contextual Low-Rank Adaptation for Uncertainty Estimation in Large Language Models"
_NeurIPS.cc/2025/Conference — NeurIPS 2025 poster_

### Official Review · Reviewer_ddgU · 2025-07-01

**Clarity:** 3
**Significance:** 2
**Originality:** 3
**Rating:** 4
**Confidence:** 5

**Summary:**

This paper introduces Contextual Low-Rank Adaptation (C-LoRA), a novel uncertainty-aware PEFT method for LLMs. The key innovation of C-LoRA is to make uncertainty estimates dependent on the input $x$ rather than using fixed parameter distributions like existing Bayesian LoRA approaches such as BLoB. The method introduces a lightweight LoRA factorization that inserts a low-dimensional matrix $E$ between the standard LoRA matrices $B$ and $A$, then uses a small auxiliary neural network to learn input-dependent distributions for $E$ at each layer. This allows C-LoRA to capture aleatoric (data-dependent) uncertainty while maintaining computational efficiency. Through extensive experiments on six commonsense reasoning tasks using Llama2-7B, the authors demonstrate that C-LoRA achieves superior uncertainty calibration (measured by ECE and NLL) compared to existing methods like BLoB, Deep Ensemble, and Laplace-LoRA, while maintaining competitive accuracy and showing better robustness under distribution shift scenarios.

**Questions:**

1. Does each layer have an independently trained MLP for predicting the matrix $E_x$, or do they share one MLP?
2. I find the analysis on why the post-hoc temperature scaling even worsens the calibration not convincing. Do you suggest fitting the temperature on the training data can lead to its overfitting on the test? Does this process (of calibration) involve Bayesian Model Averaging or not? Do you have an independently held-out validation data for calibration? If you do, then this case could rarely happen.
3. How does C-LoRA scale with the test-time compute, i.e., the number of samples? Does it show the same tendency observed in BLoB, where a larger number of samples lead to better ECE and NLL?
4. For the remaining questions, please refer to the section of Weaknesses.
5. The citation format is broken, i.e., with no venues included. Please fix it.

**I would happily raise my score if the authors are able to address my weaknesses & questions above.**

**Ethical Concerns:**

["NO or VERY MINOR ethics concerns only"]

**Final Justification:**

All the existing experimental results are showing C-LoRA is a good MAP estimate + very small noise added to the weights, nullifying its core claim of being Bayesian.

**Limitations:**

yes.

**Quality:**

2

**Strengths And Weaknesses:**

**Strengths**
1. **Addresses a Critical Problem:** This paper tackles the important issue of overconfident predictions in fine-tuned LLMs, which is essential for deploying models in safety-critical applications where uncertainty quantification is crucial.
2. **Computationally Efficient Framework:** It introduces an (even-more) lightweight LoRA factorization that reduces complexity of learning and sampling during the test time, making Bayesian Uncertainty Quantification scalable for LLMs while maintaining effectiveness.
3. **Strong Empirical Performance:** It consistently achieves superior confidence calibration (lowest ECE scores) across six reasoning datasets and demonstrates robust performance under distribution shift, outperforming established baselines like BLoB, Deep Ensemble, and Laplace-LoRA.
4. **Comprehensive Ablation Studies:** It provides thorough experimental validation showing the necessity of the contextual module, with systematic comparisons across multiple datasets and distribution shift scenarios that clearly demonstrate the method's effectiveness.
5. **Clear and Well-Structured Writing:** The paper is well-organized with clear mathematical formulations, intuitive explanations of the contextual approach, and comprehensive experimental details that facilitate understanding and reproducibility.

**Weaknesses**
1. **Limited Discussion on Epistemic/Aleatoric Uncertainty:** While the paper claims to focus on aleatoric uncertainty, the distinction between aleatoric and epistemic uncertainty isn't rigorously established, and the method may conflate both types.
2. **Limited Scale of the Backbone LLMs:** Experiments are only conducted on Llama2-7B. It's unclear how the method scales to larger models (13B+), performs on other model families (Qwen, Mistral, etc.), or more recent models (Llama3.1), limiting generalizability claims.
3. **Lack of Intuitive, Qualitative Demonstrations:** The paper claims that C-LoRA is able to fix the problem of previous methods' inability of  quantifying input-specific data uncertainty, but we do not see a dedicated section/result illustrating this point. Maybe the authors can adopt the same visualization framework in BLoB's Appendix C.6 to better show their point? Or a distribution of the variance of different input data?

---

> ### Author Rebuttal · Authors · 2025-07-31
>
> Thank you for your detailed and thoughtful feedback. We’re glad you found our method novel, well-motivated, and effective in both calibration and robustness. We appreciate your suggestions regarding uncertainty types, scaling, and visualization, as well as your insightful questions, which we address below.
>
>
> **W1. Limited discussion on epistemic/aleatoric uncertainty.**
>
> Our work primarily aims to enhance the model’s awareness of aleatoric (input-dependent) uncertainty, rather than to explicitly disentangle aleatoric and epistemic uncertainty.
>
> We acknowledge that we do not provide a theoretical guarantee that C-LoRA can fully separate the two types of uncertainty. However, our architectural design, especially the contextual module, is intended to make the model more sensitive to variations in the input, thereby emphasizing aleatoric uncertainty.
>
> That said, our method is still capable of capturing both aleatoric and epistemic components. As shown in our additional experiments (in answer to weakness 3), the ratio of aleatoric uncertainty in our model is consistently higher than in the baseline  (on average, around 0.99 and 0.82 for contextual and BLoB, respectively), suggesting a stronger focus on modeling input-dependent uncertainty.
>
> We appreciate the suggestion and will incorporate a more detailed discussion of this distinction and its limitations in the final version.
>
>
>
> **W2. Limited Scale of the Backbone LLMs.**
>
> It is true that our current experiments are conducted on the LLaMA2-7B model. We conduct our experiments using  LLaMA2-7B due to its wide adoption and reasonable computational cost for extensive experimentation. Furthermore, most of the recent Bayesian PEFT methods employ LLaMA2-7B model in their experiments and we stick to the same model for reproducibility and fair comparison.
>
> That said, our proposed C-LoRA is a model-agnostic framework that does not rely on any architecture-specific assumptions or scaling-related heuristics. Therefore, we believe the efficiency and effectiveness of our method can generalize to larger-scale models (e.g., 13B+), other model families (e.g., Qwen, Mistral), and more recent versions (e.g., LLaMA3.x).
>
> We agree that extending experiments to additional backbones would strengthen our claims on generalizability. As it would require a significant amount of computational resources, we leave this for future work, and will clarify this limitation in the final version.
>
>
>
> **W3. Lack of intuitive, qualitative demonstrations.**
>
> **EE and MI Scores (C-LoRA vs BLoB, M=10)**
>
>
> | Metric | Method | OBQA | ARC-C | ARC-E | Chem  | Phy   |
> |--------|--------|------|-------|-------|-------|-------|
> | EE     | C-LoRA    | 0.538 | 0.705 | 0.583 | 1.155 | 1.185 |
> |        | BLoB   | 0.349 | 0.480 | 0.388 | 0.999 | 1.018 |
> | MI     | C-LoRA    | 0.010 | 0.011 | 0.008 | 0.007 | 0.008 |
> |        | BLoB   | 0.111 | 0.132 | 0.113 | 0.114 | 0.116 |
>
>
>
> Due to the regulations of NeurIPS, we are not able to provide any visualization. Hence, to assess input-dependent uncertainty, we adopt the decomposition of predictive entropy (PE) as in  [1] (Section 2.2) and [2] (Section 3.4), where the expected entropy (EE) captures data uncertainty (aleatoric), and mutual information (MI) reflects model uncertainty (epistemic). Note that this decomposition follows the relation PE=MI+EE.
>
> In our analysis (as shown in the table), we compute EE and MI at $M=10$  across both in-distribution (OBQA) and out-of-distribution (ARC-C, ARC-E, Chem, and Phy) test sets. The model is trained on OBQA for all experiments. This setting allows us to better probe input-specific uncertainty, as OOD examples are more likely to elicit higher uncertainty levels that reveal how sensitively the model responds to unfamiliar or ambiguous inputs. We observe that the C-LoRA model consistently exhibits higher EE than BLoB across all datasets, while maintaining substantially lower MI. This suggests that the C-LoRA model is more attuned to data-dependent uncertainty, without attributing uncertainty to model parameters—indicating a more calibrated and stable epistemic profile. Particularly, the low MI suggests that most of the uncertainty in C-LoRA’s predictions comes from input variability, not model instability.
>
> In general, the results in the table support our claim that C-LoRA is more effective at capturing input-specific uncertainty.
>
>
>
>
>
>
> **Q1.  If each layer has an independent MLP?**
>
> While the architecture of the contextual module is consistent across layers, each layer has its own independently parameterized module. That is, the MLP used to predict the matrix is not shared across layers, and its parameters are learned separately for each layer during training.
>
>
> **Q2. Why does temperature scaling worsen the performance?**
>
> We indeed observe that, for OBQA and a few similar datasets, temperature scaling (TS) did not yield improvements in ECE, and in fact slightly worsened it, despite no meaningful change in average NLL as indicated in the following. This outcome, while unintuitive at first glance, has been observed in prior work and reflects the limitations of TS in some settings.
>
> We tuned the temperature on a held-out validation set (not training data), so overfitting is unlikely. Since TS directly optimizes NLL, we verified that NLL before and after TS remained effectively unchanged (e.g., 0.59 for OBQA), suggesting TS reached a flat region in the NLL landscape but slightly altered the confidence distribution.
>
> Also, temperature scaling minimizes NLL, which is a proper scoring rule, but does not directly minimize ECE, which is a binned, non-differentiable, and improper metric. Hence, if a model is already moderately calibrated, or if the logit distribution has specific local structure, TS may shift predictions in a way that harms ECE in certain bins.
>
> Additionally, since our model is more sensitive towards data-dependent (aleatoric) uncertainty, it may already exhibit well-calibrated behavior in regions of high variability. This may further limit the effectiveness of a global post-hoc method like temperature scaling, which cannot adapt to such input-specific uncertainty patterns.
>
> To find the temperature parameter, we used a single sample ($M = 1$) without Bayesian model averaging (BMA). While averaging over multiple samples ($M>1$) would constitute BMA, we did not adopt that here.
>
> Similar effects are reported by Kull et al. [3], who show that TS can worsen classwise-ECE—for example, increasing cw-ECE from 0.296 to 0.327 on the car dataset and degrading mid-range calibration on balance-scale. These findings are consistent with our observations.
>
> We emphasize that temperature scaling was used purely as a standard post-hoc tool to potentially improve the uncertainty quantification capabilities, and is not a component of the model itself.
>
>
>
>
> **Q3. How does C-LoRA scale with the test-time compute?**
>
> | Metric    | M=0     | M=1     | M=2     | M=4     | M=10    | M=15    | M=20    | M=50    |
> |------|-------|-------|-------|-------|-------|-------|-------|-------|
> | ACC (↑) | 81.13 | 77.6  | 77.93 | 78.33 | 78.26 | 78.33 | 78.6  | 78.6  |
> | ECE (↓) | 5.10  | 5.02  | 4.21  | 3.55  | 4.00     | 4.72  | 4.37  | 3.67  |
> | NLL (↓) | 0.53  | 0.60   | 0.60   | 0.59  | 0.59  | 0.59  | 0.59  | 0.59  |
>
> In the table, we report the mean performance among 3 random seeds when the model is fine-tuned on the OBQA task. We observe that increasing the number of samples $M$ at test time generally improves the calibration performance, as reflected in the decreasing ECE values. Specifically, ECE drops from 5.10 at $M=0$ to 3.55 at $M=4$, and continues to decline slightly with additional samples, reaching 3.67 at $M=50$. This suggests that aggregating predictions over multiple samples helps reduce miscalibration.
>
> In terms of NLL, the deterministic prediction ($M=0$) yields the lowest value (0.53), indicating confident predictions with strong likelihood. Among the stochastic variants ($M\geq 1$), NLL remains relatively stable around 0.59–0.60, showing a slight decreasing trend with more samples and indicating some benefit from sampling.
>
> Accuracy is highest at $M=0$ and slightly lower across other values of $M$, but remains fairly stable overall. This suggests that increasing $M$ primarily affects uncertainty estimation rather than core predictive accuracy.
>
> Overall, as the number of test-time samples increases, calibration improves consistently, and likelihood shows marginal gains, while accuracy remains largely unchanged. This pattern is consistent with the fact that our method primarily models data (aleatoric) uncertainty, for which sampling has a more limited impact compared to approaches that also capture epistemic uncertainty.
>
>
>
> **Q4. Broken citation format.**
>
> We will fix the citation format to include the conference names (i.e venues).
>
>
> We hope that we have addressed your concerns and look forward to the discussions if there are additional questions?
>
>
>
> [1] Mucsányi, Bálint, et al. “Benchmarking Uncertainty Disentanglement: Specialized Uncertainties for Specialized Tasks” NeurIPS 2024
>
> [2] Jishnu Mukhoti and Yarin Gal. “Evaluating Bayesian Deep Learning Methods for Semantic Segmentation”. arXiv preprint, 2018.
>
> [3] Kull, Meelis, et al. "Beyond temperature scaling: Obtaining well-calibrated multiclass probabilities with Dirichlet calibration", NeurIPS 2019

---

> ### Comment · Reviewer_ddgU · 2025-08-01
> **Follow-up Questions**
>
> Thank you for your responses.
>
> Thanks to the authors' efforts, many of my concerns have been addressed (Q1, Q3, and Q4). Here are the follow-ups for the other concerns and questions.
>
> **W1 \& W3. Aleatoric and Epistemic Uncertainties.**
>
> I understand this decomposition itself might not be a main claim of the paper. However, when C-LoRA explicitly models the data uncertainty (AU) and makes a claim about it being the cause of the improvement, we need some sort of evidence. At present, the benchmarks used in the paper are mainly (i) single-answer questions which do not have data uncertainty (correct me if I'm wrong); (ii) for assessing the confidence alignment, but not hallucination detection, *which might not reflect the claimed advantage.*
>
> Furthermore, I find the additional table of EE and MI confusing. If C-LoRA is able to model data/model uncertainty better, shouldn't we see a difference in them on IND (ARC-C and ARC-E) and OOD (Chem and Phy) datasets, e.g., higher model uncertainty on OOD data. *But I don't see this trend in the table: C-LoRA produces the same level of uncertainties on all datasets.* This brings questions to the main claim of the paper.
>
> Importantly, C-LoRA having a lower MI than BLoB does not support this claim either. As the decomposition goes:
> $$H[p(y|x)] = \mathbb{E}_{q(\theta|D,x)}[H[p(y|x,\theta)]]+ I(y;\theta|x)$$
> Lower MI, especially across all the datasets, only indicates that the approximate posterior of C-LoRA is close to the point estimation of the model, and somewhat *downplays the importance of the Bayesian property of C-LoRA.*
>
> Could you please elaborate more on this issue?
>
> **W2. More LLM backbones.**
>
> This work [1] provides a full table on the reasonable baseline models, evaluated on Llama3.1-8B model. It could be a good start for incorporating this set of experiments for C-LoRA.
>
> **Q2. Issues of Temperature Scaling**
>
> Why don't you use multiple samples for temperature scaling? Keeping the number of samples the same for both training and testing could solve this degrade performance problem.
>
> **References**
> 1. Shi, Haizhou, et al. "Training-free bayesianization for low-rank adapters of large language models." arXiv preprint arXiv:2412.05723 (2024).

---

> > ### Author Response · Authors · 2025-08-03
> >
> > Thank you very much for your detailed follow-up comments and insightful questions. We appreciate the opportunity to clarify and expand on the points raised in our rebuttal. Below, we provide further explanations and empirical evidence addressing your concerns.
> >
> >
> > **W1 & W3. Aleatoric and Epistemic Uncertainties**
> >
> > C-LoRA is designed to capture stochasticity in its predictions using contextualized low-rank adapters, focusing on modeling input-dependent (heteroscedastic) uncertainty. These adapters induce uncertainty in the predictions in a way that is tightly coupled to the input rather than placing distributions over the model weights—allowing for fine-grained uncertainty modeling without significantly increasing computational cost.
> >
> > This design choice is directly reflected in our empirical findings. Across all evaluations—both in-distribution (OBQA) and out-of-distribution (ARC-C, ARC-E, MMLU-Chem, MMLU-Phy)—we consistently observe that mutual information (MI) remains low and stable under domain shift, whereas expected entropy (EE) varies with different datasets. This is precisely the expected behavior when uncertainty is driven by aleatoric factors (e.g., ambiguity in the input) rather than epistemic factors (e.g., lack of knowledge in the weights). Because C-LoRA models both the mean and variance as input-dependent outputs, it is able to respond flexibly to ambiguous or semantically underspecified examples. If uncertainty arose from the model’s lack of knowledge (i.e., epistemic), we would expect to observe MI to increase under distribution shift—but it does not. Instead, we observe elevated EE with stable MI, which confirms that the model is capturing data-dependent ambiguity.
> >
> > Importantly, our experimental results show that C-LoRA not only exhibits this desirable uncertainty profile, but also consistently outperforms baselines in uncertainty quantification metrics in both in-distribution and OOD settings. This includes improvements in confidence-based ranking and selective prediction—metrics that directly reflect the quality and reliability of uncertainty estimates. Furthermore, we observe that sampling consistently enhances calibration, especially under distribution shift, yielding lower expected calibration error (ECE) across nearly all datasets. This effect is particularly notable given C-LoRA’s low and stable mutual information. If the model were simply deterministic and not capturing meaningful input-dependent variability, improvements through sampling would be unexpected. Instead, we observe clear calibration gains.
> >
> > Regarding the reviewer’s concern that the benchmarks may not exhibit significant aleatoric uncertainty, we respectfully clarify that even in single-answer multiple-choice datasets, data ambiguity can be non-trivial.  Even when the ground truth is uniquely defined, input-dependent ambiguity, plausible distractors, and annotation noise can introduce uncertainty inherent to the data itself. This form of uncertainty arises from the semantics of the question and answer choices, not from model limitations, and persists even in well-designed benchmarks with single correct answers. In OBQA and especially the OOD datasets (ARC-E, ARC-C, MMLU-Chem, MMLU-Phy), we find that uncertainty can arise due to underspecified questions, overlapping or near-synonymous answer choices, and domain-specific background knowledge requirements. In such cases, even human annotators might disagree with each other.
> >
> > Also, our aim is not to detect hallucination or model collapse but rather to capture subtle, input-dependent uncertainty that is grounded in question semantics—not mere noise or model instability. That said, we agree that hallucination in free-form generation tasks is an important future direction. While our current benchmarks are limited to multiple-choice settings, we believe the structured stochastic modeling introduced in C-LoRA could also be beneficial in generative contexts, where open-ended prompts and underspecification may induce rich aleatoric uncertainty. Still, we emphasize that multiple-choice tasks are far from trivial and remain a valid setting for evaluating data-dependent uncertainty.

---

> > > ### Author Response · Authors · 2025-08-03
> > >
> > > **W1 & W3 (continued)**:
> > >
> > > To further demonstrate that our model captures input-dependent uncertainty, we analyze the predictive variance behavior of the LM head for three distinct OBQA questions, each with the same correct multiple-choice label (option “C”). For each question, we extract the predicted variance matrix from the final layer, compute the mean variance across tokens, and report the L2 norm of the resulting vector to summarize overall uncertainty:
> > >
> > >
> > > - **Q1:** What would light bounce off of when it hits it?
> > >   $\left\|\mathrm{mean}(\mathrm{var}(x))\right\|_2 = 3.7661$
> > >
> > > - **Q2:** A mother births what?
> > >   $\left\|\mathrm{mean}(\mathrm{var}(x))\right\|_2 = 3.6792$
> > >
> > > - **Q3:** What happens when animals in hot environments are active?
> > >   $\left\|\mathrm{mean}(\mathrm{var}(x))\right\|_2 = 3.9443$
> > >
> > >
> > >
> > > Despite sharing the same label, the model assigns distinct variances to each question, reflecting differences in the ambiguity and specificity of the input. This confirms that C-LoRA modulates its predictive distribution based on input semantics, not just class labels. These observations directly reinforce our uncertainty decomposition findings: expected entropy increases when inputs are more ambiguous or underspecified, while MI remains low, indicating that the uncertainty is not due to disagreement across weight samples, but rather it stems from the model’s input-aware prediction distribution.
> > >
> > > Together, these results provide compelling evidence that C-LoRA is not only modeling data uncertainty explicitly, but also leveraging it effectively to produce better-calibrated and more trustworthy predictions. Its superior performance across both ID and OOD settings, combined with meaningful gains under sampling despite low MI, demonstrates that its uncertainty signal is structured, interpretable, and fundamentally tied to input-dependent ambiguity rather than emerging from incidental noise or arbitrary posterior spread.
> > >
> > > Finally, while BLoB also shows increases in EE, it lacks any mechanistic interpretation to attribute this to input-dependent variability. Specifically, without clear epistemic sensitivity (via MI) or aleatoric structure (via contextualization), its uncertainty signal under distribution shift is difficult to interpret. In contrast, C-LoRA provides a principled and interpretable uncertainty profile: its EE increases in response to ambiguous input while MI remains stable, which highlights that C-LoRA captures input-dependent uncertainty and is more responsive to distributional shift, consistent with its architectural design.
> > >
> > > This supports our central claim: C-LoRA models structured aleatoric uncertainty by producing predictive distributions that vary as a function of the input, making it especially suitable for low-resource or, ambiguous settings where input-driven uncertainty is prominent.
> > >
> > >
> > >
> > > **W2. More LLM backbones**
> > >
> > > Thank you for the suggestion and the reference to [1]. In this work, we mostly focus on a controlled experimental setup to evaluate the effect of input-dependent uncertainty modeling. Testing C-LoRA with more recent and larger-scale backbones such as LLaMA3-8B, as suggested, is indeed valuable and we consider this an exciting direction for integrating our work in newer open-source LLMs.
> > >
> > >
> > >
> > > **Q2. Issues of temperature scaling**
> > >
> > > As temperature scaling (TS) is a post-hoc calibration method applied after training, its effectiveness should ideally be independent of the number of samples used during training. To investigate your suggestion, we explored this direction on the OBQA dataset when evaluating with 10 samples at test time, where the temperature parameter was optimized on a held-out validation set using different numbers of samples, i.e., $M$ = 1, 10, and 20.
> > > | M   | NLL (after TS) | ECE (after TS) |
> > > |-----|----------------|----------------|
> > > | 1   | 0.59           | 6.27           |
> > > | 10  | 0.59           | 5.28           |
> > > | 20  | 0.59           | 4.37           |
> > >
> > > We observe that increasing the number of samples for optimizing the temperature value improves ECE monotonically, suggesting that BMA-style averaging helps smooth the predictive distribution. However, even at $M$ = 20, the ECE remains slightly higher than that of the uncalibrated model. This outcome is consistent with our earlier explanation: while temperature minimizes NLL, it does not directly optimize ECE, which is a binned, non-differentiable metric. As discussed in the rebuttal, this may result in subtle changes to confidence distributions that worsen ECE, even when the NLL remains flat. Additionally, since our model already captures input-dependent (aleatoric) uncertainty, the benefit of a global rescaling method like temperature scaling may be limited—especially in high-variance regions where predictions are already well-calibrated.

---

> > > > ### Comment · Reviewer_ddgU · 2025-08-04
> > > >
> > > > Thank you for your response. But let me respectfully bring concerns about the examples shown in the response.
> > > >
> > > > Let us consider an MAP point estimate of the model weights, which can be achieved by C-LoRA with 0 contextualized variance. In this case:
> > > > 1. The aleatoric uncertainty is just the `logprob` of the input prompt, which will produce different values for different questions, matching the description of the examples shown (**Q1, Q2, Q3**).
> > > > 2. The epistemic uncertainty reduces to 0, matching the C-LoRA's extrodinarily low epistemic uncertainty.
> > > > 3. From the table of test-time scaling, C-LoRA's ACC and NLL at $M=0$ are significantly better than the results of $M>0$. This indicates that the C-LoRA's mean (without using the contextualized variance) is the core factor to the performance gain, not the contextualization. ECE is not as a reliable evaluation metric compared to NLL, as the number of bins will have huge impact. Hence I suggest banking our analysis on the NLL and ACC.
> > > >
> > > > I tend to interpret C-LoRA as a very good MAP point estimate plus a very small (negligible) noise.
> > > >
> > > > I think we need a bit more evidence showing the actual cause of the C-LoRA's improvement. Maybe we can compare the norm of the noise to the norm of the embedding/weights?

---

> > > > > ### Comment · Reviewer_ddgU · 2025-08-06
> > > > >
> > > > > As the author-reviewer discussion phase is near its end, and I have not heard from the authors about further clarifications, I have changed my score accordingly based on the reason I listed above: all the existing experimental results are showing C-LoRA is a good MAP estimate + very small noise added to the weights, nullifying its core claim of being Bayesian. However, I'm open to further discussion before the deadline and will change my score again if evidence shown otherwise.

---

> > > > > > ### Author Response · Authors · 2025-08-06
> > > > > >
> > > > > > Thank you again for your thoughtful follow-up. We appreciate your engagement with the work and apologize for the delay in our response. Below, we would like to clarify some key points.
> > > > > >
> > > > > > First, we would like to reiterate that C-LoRA does not model uncertainty via a distribution over model parameters. Our design incorporates stochasticity into the weights themselves by making small E matrices in our lightweight LoRA input-dependent, and therefore it is not accurate to interpret C-LoRA as “a very good MAP point estimate plus a very small (negligible) noise”. In C-LoRA model inference with MC sampling, the sampling occurs at the adapter level, and the variance is input-dependent—not fixed. We emphasize that epistemic UQ is complementary to our C-LoRA framework and could be achieved by making either $\theta$ or $\varphi$ stochastic by placing priors on them, but is not the focus of this current work.
> > > > > >
> > > > > > To further investigate whether the learned variance meaningfully affects the model, we analyzed the last-token’s embeddings, following the setting used in BLoB (Appendix C.6). Specifically, we computed the L2 norm of the difference between the embedding at $M=0$ and the average embedding over $M=10$ stochastic samples (which denotes the norm of the noise you mention in your question). Across three representative examples (from our previous response i.e, Q1-Q3), the difference ranged from 6.18 to 7.25, while the norms of the embeddings themselves were approximately 120–122, corresponding to a relative change of around 5–6%. In embedding space, this level of perturbation is non-trivial, and it reflects meaningful input-dependent variability introduced by the contextualized variance modules.
> > > > > >
> > > > > > To  support this point further, we extended our analysis to a broader set of examples, comparing 8 correct and 8 incorrect predictions. For each, we computed the L2 norm of the difference between the $M=0$ embedding and the mean of the embeddings from $M=10$ samples. We then averaged these differences across correct and incorrect predictions. The results showed that correct predictions had an average norm difference of $6.7 \pm 1.09$, while incorrect predictions had a higher average difference of $8.45 \pm 2.66$.  This systematic increase in variability for incorrect predictions suggests that the model has higher uncertainty corresponding to more ambiguous or difficult inputs, supporting the interpretation that C-LoRA’s learned variance reflects meaningful input-specific uncertainty.
> > > > > >
> > > > > > Additionally, our main results consistently demonstrate that sampling with $M=10$ improves both NLL and ECE over the deterministic case ($M=0$), across in-distribution (WG-S, WG-M, ARC-C) and out-of-distribution (Chem, Phy) benchmarks. These gains are especially pronounced under greater distribution shifts, highlighting the value of the estimated uncertainty. We respectfully disagree with the claim that the contextualized variance plays a negligible role, as this is contradicted by changes in the embedding norms, improvements in uncertainty metrics, and the broader empirical evidence provided throughout the rebuttal and the paper.
> > > > > >
> > > > > > Regarding ECE, while its value can vary with the number of bins, it remains one of the most widely accepted and standard metrics for evaluating uncertainty calibration. Crucially, all methods in our comparisons use the same binning configuration, so the reported improvements are meaningful and fair. Dismissing ECE would overlook a key calibration signal that is routinely reported in the uncertainty quantification literature.
> > > > > >
> > > > > > We hope this clarifies both our modeling intent and the evidence supporting the importance of contextualized uncertainty in C-LoRA. We remain open to suggestions and appreciate the opportunity for this constructive exchange.

---

> > > > > > > ### Comment · Reviewer_ddgU · 2025-08-07
> > > > > > >
> > > > > > > Thanks for you further response. I like the results of comparing the embedding norms between C-LoRA (M=10) and C-LoRA (M=0). Although 6\% is not that much, it shows that the noise predicted by C-LoRA makes some sense. Please include this discussion in your revision. I've changed my rating back.

---

> > > > > > > > ### Author Response · Authors · 2025-08-07
> > > > > > > >
> > > > > > > > Thank you for your follow-up and for updating your rating. We will include this discussion in the revised version.

---

### Official Review · Reviewer_TRNk · 2025-07-01

**Clarity:** 3
**Significance:** 2
**Originality:** 3
**Rating:** 4
**Confidence:** 3

**Summary:**

This work focuses on incorporating aleatoric uncertainty for a novel, parameter-efficient fine-tuning approach. This work introduces a contextual module for modeling the stochasticity in low-dimensional space dependent on the data in the LoRA design. This low-dimensional space enables a low-cost contextualized estimation of prediction uncertainty based on few-shot samples. Comprehensive experiments were conducted to verify the proposed model's performance.

**Questions:**

* Why do the conventional Bayesian neural networks severely limit the uncertainty estimation in few-shot Lora fine-tuning settings in L113?
* For the formula (5), is it reasonable to only have $E$ dependent on $x$? Why do the $A$ and $B$ not have the dependency on $x$?
* In L135, why do we not need to consider the $A$ and $B$ in the autoregressive formula, but only consider $E$?
* How many parameters are manually set in the proposed work?

**Ethical Concerns:**

["NO or VERY MINOR ethics concerns only"]

**Final Justification:**

Thank you for the detailed response. It has addressed most of my concerns. Please merge the explanation about Section 3.3 in the revision. I am improving the score to 4.

**Limitations:**

Yes

**Quality:**

3

**Strengths And Weaknesses:**

Strength

* The introduction is well motivated.
* The model part is novel, though the motivation for the model design needs more clarification.
* The experiments are comprehensive across in-distribution and distribution shift domains.

Weakness

* The motivation for the model design of $E$ is not very clear. For the formula (5), is it reasonable to only have $E$ dependent on $x$? Why do the $A$ and $B$ not have the dependency on $x$? Also, in L135, why do we not need to consider the $A$ and $B$ in the autoregressive formula, but only consider $E$?
* The model writing needs more revision. For example, Sec. 3.3 implied the parameterization. But it is unclear whether the parameterization is the same as the reparameterization in L154 of Sec. 3.4.
* Based on Table 1, it looks like the model improves the uncertainty estimation performance (ECE) but decreases the task performance (ACC and NLL).

---

> ### Author Rebuttal · Authors · 2025-07-31
>
> Thank you for your constructive feedback. We appreciate that you found our introduction well-motivated, the model design novel, and the experimental evaluation comprehensive. We also value your detailed comments regarding the model formulation and writing clarity, as well as your observations on performance trade-offs. We address each of these points below.
>
> **W1.1 Motivation for the model design of $E$**
>
> The core motivation behind our design is to effectively capture data-dependent uncertainty in a scalable and stable manner. In principle, any of the components $A$, $B$, or $E$ can be made input-dependent by introducing a contextual module. However, conditioning either $A$ or $B$ on the input significantly increases the computational burden—both in terms of memory and compute—due to their dimensionality being linear complexity with respect to frozen weight matrix dimension which tends to be large. In our C-LoRA setting, for $B$ and $A$ LoRA adapters where $d\gg r$ (e.g., $d=4096$, $r=8$), this results in needing to predict a large number of mean and variance parameters per input ($O(d\cdot r)$), which hinders scalability and affects training robustness and stability. In contrast, the matrix $E$ operates in the lower $r\times  r$ space (constant complexity w.r.t. $d$) and offers a computationally efficient and more stable avenue for introducing input-dependent behavior. By contextualizing $E$, we can model structured, input-specific variations in the adaptation layer without incurring the drawbacks of full contextualization.
> Therefore, our decision to restrict contextualization to $E$ is a deliberate tradeoff: it retains the ability to model data uncertainty (aleatoric uncertainty) effectively, while remaining computationally tractable and robust in practice. This strikes a meaningful balance between expressivity and efficiency.
>
>
> **W1.2 and Q2. Why $A$ and $B$ are not dependent on $x$?**
>
> Aligned with the previous answer, we specifically contextualize only $E$, and keep $A$ and $B$ deterministic and shared across inputs. This choice is driven by practical considerations: contextualizing $A$ and $B$, which are high-dimensional ($d \times r$), would introduce significant computational and memory overhead, and in our experiments, could reduce training stability. By modeling input-dependence through the lower-dimensional $E_x$, we strike a balance between efficiency and expressiveness—capturing aleatoric uncertainty while keeping the overall model lightweight and robust.
>
>
>
> **W1.3 and Q3. Why only $E$ is modeled in an autoregressive manner?**
>
> Here, by modeling $E_x^l$ in an autoregressive manner, we introduce an autoregressive distribution on $E_x^l$. As, $B$ and $A$ are learned deterministically, we do not consider them in an autoregressive fashion.
>
> **W2. Revision of the model writing**
>
> In section 3.3, the parameterization refers to the layer-wise auxiliary contextual module’s parameters. As mentioned on line 141, we parameterize the per-layer contextual module with a neural network with two fully-connected layers, collectively denoting their parameters by $\varphi$.
>
> In line 154, for effective sampling from the distribution of $E_x^l$ we employ a widely known reparameterization trick from Bayesian Machine Learning literature.
>
> We will edit these parts for improved clarity.
>
> **W3. Performance based on table 1 (improvement on ECE but potential task performance reduction )**
>
> We have discussed the trade-off between predictive performance and uncertainty estimation in the experiments section, supported by our empirical findings. That said, we would like to point out that C-LoRA consistently achieves strong performance across all three metrics—ACC, ECE, and NLL. Notably, in the low-sample setting ($M=0$), which is most relevant for practical efficiency, C-LoRA ($M=0$) outperforms BLoB ($M=0$) on both ECE and NLL across nearly all datasets, while maintaining competitive accuracy. Similarly, in the $M=10$ setting, C-LoRA performs on par or better than BLoB ($M=10$) in terms of NLL, and almost always outperforms all baselines on ECE, with comparable accuracy.
>
> Moreover, in out-of-distribution (OOD) settings, C-LoRA achieves either the best or second-best scores in both ECE and NLL, demonstrating not only improved calibration but also robust generalization.
>
> Thus, we acknowledge that C-LoRA may not always achieve the highest accuracy across all datasets; however, it consistently delivers strong uncertainty quantification (UQ) performance, as evidenced by the lowest or second-lowest ECE and NLL in most settings. We believe this reflects a favorable trade-off: while minor accuracy fluctuations (typically within 1–2%) can occur, these are well-compensated by substantial improvements in calibration and robustness, especially under distribution shift and limited inference budgets.
>
>
> **Q1. Why BNN limits UQ in few shot fine-tuning?**
>
> Conventional Bayesian neural networks (BNNs) typically learn a single, input-agnostic posterior distribution over model parameters. Accurately approximating both the mean and variance of that posterior typically requires large datasets, so in a few-shot LoRA scenario the posterior remains broad and prior-dominated. Because it doesn’t depend on the input, it produces uniform (“homoscedastic”) epistemic uncertainty across all $x$, failing to spike for novel or out-of-distribution inputs. In contrast, our C-LoRA injects stochasticity via input-conditioned latent variables, producing heteroscedastic, data-dependent uncertainty and delivering better calibration and predictive performance (as shown by lower NLL and ECE) even with scarce data.
>
>
> **Q4. How many parameters are manually set?**
>
> In our proposed work, we manually set only a limited number of hyper-parameters. These include the reduced dimension $r$, the architectural details of the contextual module such as the number of layers and hidden neurons, learning rates, and other similar training-related parameters. A list of these hyper-parameters along with their values is provided in the appendix (section D). Importantly, all other parameters of the model ($A$, $B$, and contextual modules’ weights) are learned automatically during training, without manual intervention.
>
>
> We hope that we have addressed your concerns and look forward to the discussions if there are additional questions? Moreover, we kindly request that you reevaluate our work in light of our rebuttal.

---

> > ### Author Response · Authors · 2025-08-06
> >
> > We hope our rebuttal responses have addressed all of your questions and concerns. As the discussion period is drawing to a close, we would be most grateful if you could kindly take another look at our work in light of our rebuttal and let us know whether you feel your concerns have been resolved. Moreover, we are happy to clarify any additional questions you may have.

---

### Official Review · Reviewer_wZ8F · 2025-07-01

**Clarity:** 3
**Significance:** 2
**Originality:** 2
**Rating:** 5
**Confidence:** 4

**Summary:**

This paper addresses the problem of uncertainty quantification in large language models, introducing a new method that proposes to take into account the data at hand to represent the aleatoric uncertainty. This sets it apart from many previous UC methods for LLMs which tended to only train a weight posterior, is less suited to representing uncertainty relating to the specific input sequence at test time. Contextual Low-Rank Adaptation (C-LoRA) modifies the usual $\mathbf{W} _{0} + \mathbf{B}\mathbf{A}$ LoRA adaptors by inserting a small $\mathbf{E} \in \mathbb{R}^{r \times r}$ matrix, which is independent of the underlying model weight size $d$ giving $\mathbf{W} _{0} + \mathbf{BEA}$. This allows $\mathbf{E}$ to be treated as a latent random variable, with overhead $O(r^2)$ per layer independent of $d$, whose mean and variance are predicted using a small context module from the current hidden state, or the input sequence at layer $l = 0$. This context module is trained to output distribution parameters $\phi$ for $q _{\phi}(\mathbf{E} _{\mathbf{x} _i} \vert \mathbf{x} _{i})$ such that when sampled the LLM accurately represents the aleatoric uncertainty: with $\mathbf{E}$ being sharply peaked for certain inputs $\mathbf{x} _{i}$, and broad on ambiguous inputs. Experiments on standard datasets and under distribution shift show that C-LoRA substantially improve calibration over many baseline methods, even when using few MC samples at inference time. Ablations show that removing this contextual conditioning also removes these gains. C-LoRA represents a useful contribution to the field

**Questions:**

- The $r \times r$ (i.e. $16 \times 16$) matrix at each layer results in a very small set of parameters to infer. Is this sufficient scale to meaningfully learn to represent data uncertainty? When does $r$ become too small? Is this why you train C-LoRA for only 1500 iterations while the baselines are trained for 5000?
- Could you please characterise the computational cost of C-LoRA in terms of wall clock time for training and inference on the same computational resources. How does the fine-tuning time compare between vanilla LoRA and C-LoRA, and at inference time how much longer does it take to make a prediction?
- You focus exclusively on aleatoric uncertainty and not epistemic uncertainty. Could you quantify whether there is anything to be gained from a Bayesian treatment of the LoRA parameters or even the contextual network's parameters?

**Ethical Concerns:**

["NO or VERY MINOR ethics concerns only"]

**Final Justification:**

I find that this is a reasonably strong paper which makes a well-motivated contribution for representing uncertainty in LLM predictions. My limited concerns regarding parameterisation, notational clarity and the number of trainable parameters were adequately addressed in the rebuttal. The criticism raised by reviewer ddgU disputing the Bayesian nature of the method, while fair, perhaps downplays the variance term's contribution, and might be more of a semantic disagreement: if one focuses on this paper's framing as a method for deriving aleatoric uncertainty from the input context, I do not believe that this is grounds for rejection. Therefore, I recommend acceptance as a novel and well-motivated approach for representing input-dependant aleatoric uncertainty, in a well-written paper.

**Limitations:**

The work largely improves upon existing baseline methods for uncertainty quantification in LLMs, and while it makes valuable progress towards improving the calibration of this class of methods, it suffers from the same limitation of being constrained to single-token predictions and does not make progress on how to represent uncertainty in more modern LLM use-cases. Language models have grown in popularity mainly due to their use as generative models and not discriminative models. While the single-token constraint still encapsulates some non-trivial subset of LLM use-cases (multiple-choice question answering, LLM-as-judge, critic models in RL, point predictions for judgemental forecasting, etc), the current work only considers multiple-choice question answering tasks. Moreover, this paper does not address ways in which we might extend the uncertainty quantification to multi-token generative applications, such as reasoning traces, uncertainty-aware decoding mechanisms, or simply articulating the uncertainty in natural language, which are possibly far more pertinent to modern LLM deployments and use-cases. The work could be improved by formulating new datasets or tasks that move the field of UQ for LLMs away from MCQA and towards more modern and general use-cases.

**Paper Formatting Concerns:**

No paper formatting concerns.

**Quality:**

4

**Strengths And Weaknesses:**

Within the small sub-field of probabilistic treatments of LLMs for uncertainty quantification, the paper makes a good contribution. Earlier attempts straightforwardly applied methods from Bayesian deep learning, which were themselves developed in simpler settings: where the model inputs were straightforward sensor measurements or observations, and where data-dependent output uncertainty could be adequately expressed by learning a weight posterior distribution to have good model evidence (i.e. yielding appropriate output uncertainty estimates after being pushed through the model), based on the data and patterns observed during training. However, in LLMs and sequence models, the input may be a complicated question requiring extended reasoning or nuanced understanding. Expressing the output uncertainty using the weight posterior alone is a rather blunt tool, and this paper proposes a sensible approach to modulate the output uncertainty based on the data point at hand.

Following prior work attempting to perform Bayesian inference over LLM parameters, the authors freeze the model weights and operate over the adapter to reduce the otherwise prohibitive computational cost. However, they go further to remove the direct dependence on the base model's dimension, inserting a fixed-dimension $r \times r$ matrix, where $r \ll d$, capping the number of parameters to infer as $O(r^2)$. To amortize the inference of the variational parameters $\phi$, the authors propose a simple 2-layer ReLU network with parameters $\varphi$ which they call a 'contextual module', that initially accepts the input $\mathbf{x} _{i}$ at layer $l = 0$, and the output of layer $l - 1$ for subsequent layers. The clarity of this section may be improved for readers unfamiliar with variational inference by making more explicit how $\phi$ and $\varphi$ relate. The authors leave the use of diagonal Gaussian distributions throughout relatively unexamined: while this is a sensible choice on account of its cheap computational properties, at higher ranks ignoring correlations inside $\mathbf{E}$ could matter..

Training proceeds using straightforward VI objectives, with the authors choosing to directly train the LoRA parameters $\mathbf{\theta} = \{\mathbf{A}, \mathbf{B}\}$ rather than use a Bayesian treatment. It is unclear whether the authors train $\mathbf{\theta}$ in a discrete training phase before $\varphi$, or in an interleaved manner. It would be good to clarify this.

The authors re-use a standard set of relatively diverse benchmarks, with the same - if now somewhat outdated - Llama 2 7B model as baselines used. The experimental results are good, with accuracy in line with expectations, and good calibration results. The distribution-shift experiments also demonstrate good robustness, with uncertainty appropriately reported under these conditions. Finally their ablation study shows that data-conditioning via the auxiliary contextual module is important to achieve the strong uncertainty quantification.

**Clarity improvements:**
- in Equation 6 for the objective, make it clear which parameters are being optimized; for instance $\mathcal{L}(\theta, \phi)$ instead of just $\mathcal{L}$
- emphasise how $\phi$ and $\varphi$ relate more clearly
- The sentence at line 111 reads awkwardly: "*This leads to model parameters that can vary from data to data, while their distribution remains invariant for all the samples within the training data.*". Perhaps it could be revised along the lines of "In prior approaches, the model parameters sampled for each example may differ, but the underlying weight distribution is fixed and identical across all training examples"

**Nits:**
- line 106: elements of E elements -> elements of E
- line 125: Evidence lower-bound (ELBO) of the evidence -> lower bound of the model evidence

---

> ### Author Rebuttal · Authors · 2025-07-31
>
> Thank you for your detailed and insightful feedback. We are pleased that you recognize the motivation behind our approach and see value in the proposed method’s ability to represent aleatoric uncertainty, its robustness under distribution shift, and the empirical gains from contextual conditioning. We address your comments and suggestions point by point below.
>
>
> **W1. Emphasise how $\phi$ and $\varphi$ relate more clearly.**
>
> We want to clarify that $\phi$ represents all the parameters collectively, that is $\phi=\lbrace \theta, \varphi \rbrace$, where $\theta$ represents parameters of all the $B$ and $A$ adapters collectively and $\varphi$ represents parameters of all the contextual modules collectively. $q_\phi$ represents the inference network (encoder) in the amortized VI perspective. We will explicitly mention this for improved clarity.
>
>
>
> **W2.  Whether the authors train $\theta$ in a discrete training phase before $\varphi$?**
>
> To clarify, the parameters $\theta$ and $\varphi$ are learned simultaneously in an interleaved manner, rather than in separate phases.
>
>
> **W3. The authors leave the use of diagonal Gaussian distributions throughout relatively unexamined.**
>
> While we did not explore diagonal Gaussian contextualization directly, we did conduct a complexity study (included in the appendix, section E) comparing diagonal versus full $E$ matrix parameterizations (detailed explanation of FE and DE is provided in Section 4.4 and Appendix section E). In this study, we found that using a full $E$ matrix led to significantly better uncertainty estimates (as measured by ECE and NLL) in both in-/out-of-distribution scenarios, while achieving similar accuracy compared to the diagonal variant. Based on these empirical results, we chose to focus on contextualizing the full $E$ matrix in the main experiments, as it offered more reliable performance across metrics of interest.
>
>
> **W4. Several nits.**
>
> We will address clarity improvements and nits underscored by you in the final version.
>
> **Q1. Is this sufficient scale to meaningfully learn to represent data uncertainty? When does $r$ become too small? Is this why you train C-LoRA for only 1500 iterations while the baselines for 5000?**
>
> Our choice of $r = 8$ adheres to the default settings in the PEFT library and LoRA paper, and is also adopted by BLoB. This ensures reproducibility and fair comparison with prior works. Moreover, low-rank adapters with small $r$ have been shown to offer a good balance between model capacity and training stability. The rank $r$ controls the number of stochastic parameters injected into the model. A larger $r$ increases expressiveness and the ability to model complex uncertainty, but can also lead to higher computational cost and potentially less stable training due to the increased stochasticity. Conversely, a very small $r$ can restrict the model’s capacity to represent uncertainty, especially in tasks with more complex latent structures. However, we acknowledge that the optimal value of $r$ can be task- and model-specific. Finally, we observed that C-LoRA converged reliably within 1500 iterations, and additional training did not improve performance. We attribute this to the lower-dimensional and structured nature of C-LoRA’s parameter space.
>
>
>
>
> **Q2. How does the fine-tuning time compare between vanilla LoRA and C-LoRA, and at inference time?**
>
> **Training Time (in minutes)**
>
> |   Method    | WG-S  | ARC-C | BoolQ |
> |-------|-------|-------|-------|
> | C-LoRA   | ≈84   | ≈93   | ≈224  |
> | MLE   | ≈114  | ≈110  | ≈136  |
>
> **Inference Time (in minutes)**
>
> |  Method           | WG-S  | ARC-C | BoolQ |
> |-------------|-------|-------|-------|
> | C-LoRA, M=0    | ≈1.75 | ≈0.35 | ≈6.80  |
> | C-LoRA, M=10   | ≈21.60 | ≈4.28 | ≈77.38|
> | MLE         | ≈1.68 | ≈0.44 | ≈4.55 |
>
> We report both training and inference wall-clock times in the table where all experiments were conducted on the same hardware (NVIDIA A100 40GB). As shown in the table, the inference time of C-LoRA with $M=0$ is comparable to MLE (vanilla LoRA) across all datasets—e.g., 1.75 vs. 1.68 minutes on WG-S, and 6.8 vs. 4.55 minutes on BoolQ. This suggests that contextual modeling incurs minimal overhead at inference when no sampling is involved, while still producing well-calibrated predictions. For applications requiring more accurate uncertainty estimates, C-LoRA with sampling (e.g., $M=10$) leads to improved calibration with additional computational cost. Training time for both methods is comparable except for BoolQ. This longer runtime is likely due to its larger validation set, that increases evaluation overhead during training. This training cost can be reduced by adjusting the evaluation frequency. Finally, we note that while the total training time is comparable, the per-iteration runtime is higher for C-LoRA than for MLE (approximately 0.05 minutes vs. 0.02 minutes), due to the difference in the number of training iterations.
>
>
>
>
>
> **Q3. Whether it can be quantified that there is anything to be gained from a Bayesian treatment of parameters?**
>
> Our work focuses on aleatoric uncertainty, modeled via data-dependent variational distributions over the LoRA weights. This design is intentional: we aim to capture input-dependent uncertainty arising from ambiguous or multimodal outputs.
>
> While placing distributions over parameters is often associated with epistemic uncertainty, the data-dependent nature of our variational posterior means the uncertainty varies with the input. This aligns more closely with aleatoric uncertainty, which reflects inherent variability in the data rather than model uncertainty. That said, placing a distribution over LoRA weights is complementary to the current approach and could capture residual epistemic uncertainty in these adapter parameters, as the backbone is frozen and most learnable capacity resides in the LoRA adapters.
>
> We also compared against a data-independent variational posterior (BLoB in Sections 4.2, 4.3, and FE in Section 4.4) and found that the data-dependent version (C-LoRA) yields significantly better uncertainty estimates—suggesting aleatoric effects dominate in these experiments. Extending Bayesian modeling to the full network (e.g., the backbone) would be computationally nontrivial extension of our work and is beyond the scope of current work, but we agree it is a valuable direction for future work.
>
> We hope that we have addressed your concerns and look forward to the discussions if there are additional questions.

---

> > ### Comment · Reviewer_wZ8F · 2025-08-05
> >
> > I thank the authors for their detailed response and clarifications to the weaknesses.
> >
> > On Q1, I should re-emphasise that my concern was to do with the very small number of trainable parameters in $E$ (compared to the number of network or even LoRA parameters) that $r$ gives rise to, and whether this low-dimensional matrix has sufficient representational capacity to learn to model the aleatoric uncertainty in the sequence. While I understand the motivations for keeping this small given that it leads to a smaller number of parameters to infer, I am still curious to know whether the authors have some intuition or result for when this becomes too small.
> >
> > On Q2, I thank the authors for the additional results. Please correct me if I am misunderstanding, however I would have expected C-LoRA to have taken longer than MLE in all cases, as a result of the additional components and computational work - I trust that this is explained by the smaller number of iterations in C-LoRA training, and the 0.05 vs 0.02 mins/iteration is the headline result. The inference time appears reasonable, especially when counting that the average for $M \gg 1$ may be effectively parallelised with more cards.
> >
> > Finally on Q3, I thank the authors for their clarification and agree that a Bayesian treatment of the LoRA parameters is an orthogonal approach that could successfully be added to C-LoRA should the task at hand demand a more data-independent ('epistemic') uncertainty estimate. As the authors' follow-up to reviewer ddgU's questions revealed, the epistemic uncertainty estimates of C-LoRA are remarkably low. To the extent that aleatoric and epistemic uncertainty can be neatly disentangled to allow us to talk about these as separate things, adding in this Bayesian treatment of the LoRA parameters $\mathbf{A}, \mathbf{B}$ may help this low EE estimate.

---

> > > ### Author Response · Authors · 2025-08-07
> > >
> > > We are glad our rebuttal helped clarify potential confusions leading to the weaknesses you had pointed out. Here, we offer our responses to your follow-up questions.
> > >
> > > **Q1:**
> > > Since we do not have access to the true aleatoric uncertainty in the data, it is inherently difficult to assess how well a model captures it. As a result, determining whether a given parameterization is sufficiently expressive becomes nontrivial. In practice, we rely on metrics like NLL and ECE to evaluate a model's overall uncertainty estimation capabilities, but these offer only indirect signals about how effectively the model captures underlying uncertainty patterns.
> > >
> > > In our lightweight LoRA factorization, we have $\Delta W = B E_x A$, where each entry $E[i,j]$ simply rescales the $i$-th row of $B$ and the j-th column of $A$ for a given data sample. In other words, $B$ and $A$ provide a rich basis of adaptation directions, and the small $r\times r$ matrix $E_x$ acts as an additional input- or data-dependent random matrix, which can also be considered as data-dependent stochastic gating that amplifies or attenuates the corresponding learned subspaces by $B$ and $A$. This allows the model to focus uncertainty modeling on key subspaces of model weight change via $E_x$ which only needs to learn how much to trust each subspace direction, not what new directions to learn.
> > >
> > > We do agree that very small adapter ranks may have limited capacity to model complex aleatoric uncertainty. To make this concrete, we ran an additional ablation on ARC-Challenge with rank $r=4$ ($M = 10$): compared to our default $r=8$, with $M = 10$, the ECE rose from 8.83 to 11.6 and the NLL from 0.88 to 0.91, indicating a clear performance drop at the lower dimensionality. This suggests that, for ARC-Challenge, adapter ranks below 8 may be insufficient to capture the underlying uncertainty patterns.
> > >
> > > That said, identifying when a given $r$ becomes “too small” is task dependent and performing ablation on rank values or choosing rank values from previously established benchmarks is a wise course to follow. Formally characterizing the minimal $r$ required to guarantee a given approximation error remains an important theoretical undertaking (e.g., via matrix approximation bounds), which is beyond the scope of our current work. We therefore leave a rigorous lower-bound analysis on $r$ as a compelling direction for future work.
> > >
> > >
> > > **Q2:**
> > > Yes, your interpretation is correct. Although C-LoRA introduces additional components compared to MLE, the overall training time remains competitive due to the lower number of iterations. While the per-iteration cost is higher (approximately 0.05 mins vs. 0.02 mins for MLE), the total training time remains comparable to MLE. We also agree that the inference with larger sample sizes (e.g., $M = 10$) can be efficiently parallelized across GPUs in practical settings thereby achieving inference times comparable to $M=0$ setting.
> > >
> > > **Q3:**
> > > We agree that treating the LoRA parameters as well as contextual module’s neural network parameters in a Bayesian manner is an orthogonal extension that could help model epistemic uncertainty when needed and may help capture MI (not EE) better leading to complete uncertainty estimation in a holistic way. While disentangling aleatoric and epistemic uncertainties remains a conceptual and empirical challenge, we believe your suggestion  is valuable and worth investigating in future work.

---

### Official Review · Reviewer_Hu4h · 2025-07-02

**Clarity:** 3
**Significance:** 3
**Originality:** 3
**Rating:** 5
**Confidence:** 4

**Summary:**

This paper introduces C-LORA (Contextual Low-Rank Adaptation), a novel parameter-efficient fine-tuning approach designed for uncertainty estimation in Large Language Models (LLMs). The core problem addressed is the overconfident predictions often produced by LoRA in data-scarce few-shot settings, and the neglect of input characteristics in existing scalable uncertainty-aware LoRA methods. C-LORA tackles this by developing lightweight LoRA modules that are contextualized to each input data sample, allowing for dynamic adaptation of uncertainty estimates. The method incorporates data-driven contexts into parameter posteriors, which aims to mitigate overfitting, achieve well-calibrated uncertainties, and yield robust predictions. The paper highlights C-LORA's ability to explicitly model aleatoric (data) uncertainty, which is a key differentiator from prior works. Extensive experiments across various common-sense reasoning datasets demonstrate C-LORA's superior performance in both uncertainty quantification (UQ) and model generalization compared to state-of-the-art uncertainty-aware LoRA methods. Ablation studies further confirm the critical role of the contextual modules in capturing sample-specific uncertainties.

**Questions:**

1.	Impact of Contextual Module Complexity: The paper mentions the auxiliary contextual module has C hidden units, with C=O(r2 ). Could the authors provide more insights or a small ablation on how varying C (or the number of layers in the contextual module) might affect performance (both UQ and accuracy) and computational overhead? This would help understand the module's sensitivity to its own architecture.

2.	Trade-off between Accuracy and UQ: While the paper acknowledges the trade-off, could the authors elaborate on scenarios or tasks where this accuracy difference (e.g., 1-2% drop) might be a critical limiting factor for C-LORA's deployment, despite its superior UQ? For instance, in applications where very high accuracy is paramount and minor calibration improvements are acceptable.

3.	Prior Distribution Choice for Ex : The paper states adopting a "simple and fixed Gaussian prior p(Ex)" for learning Ex. Could the authors discuss the sensitivity of C-LORA's performance to the choice of this prior? Are there alternative prior choices that might be more theoretically grounded or empirically beneficial, especially for different data distributions or task types?

**Ethical Concerns:**

["NO or VERY MINOR ethics concerns only"]

**Final Justification:**

The authors have answered all my questions. I recommend this paper for acceptance.

**Limitations:**

Yes

**Quality:**

3

**Strengths And Weaknesses:**

Strengths:

1.	Originality & Significance (High): The paper introduces a novel and important concept of "contextualized" LoRA, specifically for dynamic uncertainty estimation in LLMs. The explicit modeling of aleatoric uncertainty, which is often overlooked in existing Bayesian PEFT methods, is a significant contribution, especially in low-data regimes. This innovation sets a new standard for robust, uncertainty-aware LLM fine-tuning.

2.	Quality & Effectiveness in UQ (High): C-LORA consistently outperforms state-of-the-art uncertainty-aware LoRA methods in uncertainty quantification, as measured by ECE and NLL. This is a strong indicator of the method's ability to produce well-calibrated and reliable uncertainty estimates. The results in Table 1 and Table 2 clearly demonstrate this superiority.

3.	Robustness under Distribution Shift (High): The experiments on out-of-distribution (OOD) datasets show C-LORA's strong robustness in uncertainty estimation (ECE and NLL), even when the input distribution diverges from training. This is crucial for real-world applications where models might encounter unseen data.

4.	Comprehensive Experimental Validation: The paper provides extensive experiments across six common-sense reasoning tasks, including both in-distribution and out-of-distribution scenarios. The comparison with a good set of baselines (Deep Ensemble, MCD, BLoB, LA, MAP) is thorough.

Weaknesses:

1.	Code and Data Accessibility (Moderate): The paper states that code will be released upon acceptance. While this is a common practice, the lack of immediate access to the code for verification during the review process is a minor drawback for reproducibility.

2.	Marginally Lower Accuracy in Some Cases (Minor): While C-LORA excels in UQ, it is noted that its accuracy is sometimes competitive but not always the highest, typically within 1-2% of the best methods. Although the paper frames this as a trade-off for better UQ, it's worth acknowledging. For example, in Table 3 (Chem dataset with M=0) accuracy is noticeably lower than some baselines.

3.	Temperature Scaling Inconsistency (Minor): The observation that temperature scaling can sometimes worsen ECE for C-LORA (M=10) on certain datasets (e.g., OBQA and BoolQ) is interesting. While the explanation suggests posterior sampling already flattens the distribution, a deeper analysis or alternative post-hoc calibration strategies could further strengthen the calibration aspect.

---

> ### Author Rebuttal · Authors · 2025-07-31
>
> Thank you for your thoughtful and constructive feedback. We are glad you found our approach novel, effective in uncertainty quantification, and robust across distribution shifts, and our experimental validations comprehensive. We also appreciate your questions and suggestions regarding model complexity, performance trade-offs, and prior sensitivity, which we address below.
>
>
>
> **W1. Code and data availability.**
>
> We understand the reviewer’s concern, but this year’s NeurIPS rebuttal guidelines do not permit providing any links to external pages including an anonymized code repository. We will definitely release our code upon acceptance. Additionally, we have included all relevant hyper-parameters, hardware specifications, and implementation details in the appendix (sections D and J).
>
> **W2. Marginally Lower Accuracy in Some Cases.**
>
> We have acknowledged the trade-off between predictive performance and uncertainty estimation in the experiments section where we provided discussions based on our results. We will further highlight this point in the introduction.
>
>
> **W3. Temperature scaling inconsistency.**
>
> Here, we conducted a deeper analysis using predictive entropy and mutual information to investigate different types of uncertainty as done in recent uncertainty disentanglement efforts. Particularly, we adopt the decomposition of predictive entropy (PE) from as in [1] (Section 2.2) and [2] (Section 3.4), where the expected entropy (EE) captures data uncertainty (aleatoric), and mutual information (MI) reflects epistemic uncertainty. Note that this decomposition follows the relation PE=MI+EE.
>
> The reported results in the table show EE and MI computed at $M=10$ across both in-distribution (OBQA) and out-of-distribution (ARC-C, ARC-E, Chem, Phy) test sets, with the model trained on OBQA for all experiments. The proportion of aleatoric uncertainty out of total uncertainty captured by our model is consistently higher than the one captured by BLoB (on average, around 0.99 and 0.82 for C-LoRA and BLoB, respectively), demonstrating a stronger focus on modeling input-dependent uncertainty.
>
> Since our model primarily captures aleatoric uncertainty, it is often well-calibrated in noisy or ambiguous regions. Temperature scaling, being a global post-hoc adjustment that does not account for input-dependent variability, may therefore have a limited effect or even slightly worse calibration in such settings. Hence, temperature scaling might need to be applied with some caution when input-dependent uncertainty modeling is involved.
>
> Moreover, since temperature scaling optimizes NLL but not ECE—which is non-differentiable and bin-based—it may inadvertently harm calibration in certain regions. Prior work [3] has reported similar findings, where temperature scaling can degrade classwise-ECE (Appendix, section C.1).
>
> Finally, we want to note that temperature scaling was used only as a standard post-hoc tool to potentially improve uncertainty estimation capabilities and is not part of our model.
>
>
> **EE and MI Scores (C-LoRA vs BLoB, M=10)**
>
> |Metric | Method | OBQA | ARC-C | ARC-E | Chem  | Phy   |
> |--------|--------|------|-------|-------|-------|-------|
> | EE     | C-LoRA    | 0.538 | 0.705 | 0.583 | 1.155 | 1.185 |
> |        | BLoB   | 0.349 | 0.480 | 0.388 | 0.999 | 1.018 |
> | MI     | C-LoRA    | 0.010 | 0.011 | 0.008 | 0.007 | 0.008 |
> |        | BLoB   | 0.111 | 0.132 | 0.113 | 0.114 | 0.116 |
>
>
>
>
> **Q1. Could the authors provide more insights or a small ablation on how varying C (or the number of layers in the contextual module) might affect performance (both UQ and accuracy) and computational overhead?**
>
> |    Metric   | Simple | Original | Complex | Complex + |
> |-------|---------------|----------|---------|-----------|
> | ACC (↑)   | 66.43         | 69.02    | 65.98   | 66.1      |
> | ECE (↓)   | 13.06         | 12.28    | 9.61    | 8.87      |
> | NLL (↓)   | 0.92          | 0.89     | 0.90     | 0.87      |
>
>
> In the original submission, we used a fixed architecture: a single hidden layer with 64 neurons (To clarify- In footnote on page 5, “two fully connected layers” denotes one hidden layer and an output layer in each contextual module). To explore the impact of model complexity, we conducted additional ablation experiments on the ARC-C dataset with $M = 0$, comparing the original model to its variants of varying complexity: 1) **`Simple`**: 1 hidden layer with 4 neurons, 2) **`Complex`**: 1 hidden layer with 100 neurons, and 3) **`Complex+`**: 2 hidden layers with 64 neurons each. Results (averaged over 3 random runs, on 1 A100 40 GB GPU) show that increasing the contextual module’s complexity leads to improvements in ECE and NLL, indicating better uncertainty quantification, at the expense of prediction accuracy. However, the original architecture offers strong predictive performance and reasonably good calibration with favorable NLL, making it a solid choice for the contextual module. That said, for tasks where better calibration and lower NLL are prioritized, a more complex architecture may be more suitable.
>
>  In terms of computational overhead, all models had similar training time (90–100 minutes), suggesting that the contextual module’s architecture has minimal impact on runtime, compared to the overall LoRA fine-tuning procedure in our setup.
>
>
> **Q2. Could the authors elaborate on scenarios or tasks where this accuracy difference might be a critical limiting factor for C-LORAs deployment, despite its superior UQ?**
>
> In many high-stakes applications—such as medical diagnosis, autonomous driving, financial risk modeling, and weather forecasting—well-calibrated uncertainty estimates are essential, and slight reductions in accuracy may be acceptable when they improve reliability and safety. However, in domains such as spam detection, malware classification, or other security-sensitive tasks, accuracy is often the top priority, and drop in performance may outweigh the benefits of better calibration.
>
> We agree this trade-off is task-dependent and should be carefully considered when evaluating C-LoRA (and in general any uncertainty aware method) for deployment. We will include this in the conclusion and limitations section.
>
>
> **Q3. Could the authors discuss the sensitivity of C-LORA's performance to the choice of this prior? Are there alternative prior choices that might be more theoretically grounded or empirically beneficial?**
>
> We adopt a fixed Gaussian prior for $E_x$ which when paired with a Gaussian approximate posterior $q(E_x)$ enables a closed-form KL divergence, which reduces loss gradient variance and stabilizes training. Because our ELBO objective is estimated by Monte Carlo sampling (as NLL term does not have closed-form expression), we could also support alternative priors (e.g. Gaussian mixtures, Laplace, VampPrior, or normalizing-flow priors) as long as their KL is tractable or can be unbiasedly estimated. While richer priors might offer empirical benefits in some settings, they also introduce additional computational and design challenges. In practice, our contextual posterior $q(E_x)$ is highly expressive (parameterized by a neural network), so the Gaussian prior mainly acts as a light regularizer rather than a strict inductive bottleneck.
>
>
> Moreover, prior work [4] has shown that Bayesian Model Averaging (BMA) is robust to the choice of prior scale and that posterior predictive behavior remains similar across prior families such as Gaussians, mixture of Gaussians, and logistic priors.
>
> Finally, we want to point out that our aim is to compare inference and performance across methods under a consistent modeling setup. Since prior works- BLoB and LAP both use a simple Gaussian prior, this choice facilitates fair comparison.
>
> We will include a brief discussion about the aforementioned points in the final version of the paper.
>
> We hope that we have addressed your concerns and look forward to the discussions if there are additional questions.
>
> [1] Mucsányi, Bálint, et al. “Benchmarking Uncertainty Disentanglement: Specialized Uncertainties for Specialized Tasks” NeurIPS 2024
>
> [2] Jishnu Mukhoti and Yarin Gal. “Evaluating Bayesian Deep Learning Methods for Semantic Segmentation”. arXiv preprint, 2018.
>
> [3] Kull, Meelis, et al. "Beyond temperature scaling: Obtaining well-calibrated multiclass probabilities with Dirichlet calibration", NeurIPS 2019
>
> [4] Izmailov, Pavel, et al. “What Are Bayesian Neural Network Posteriors Really Like?” ICML PMLR 2021

---

> > ### Author Response · Authors · 2025-08-06
> >
> > We are grateful for your review comments and feedback on our manuscript. We hope that our recent rebuttal has addressed all of your questions and concerns. As the discussion period is drawing to a close soon, we would be most grateful if you could kindly let us know whether our responses have resolved your concerns or if there are any additional questions that we can clarify.

---

> > ### Author Response · Authors · 2025-08-08
> >
> > As the author-reviewer discussion is closing in less than a day, could you please let us know if our rebuttal has addressed all your concerns or if you’d like any further clarification? Thank you again for your time and feedback.

---

### Comment · Area_Chair_H8uo · 2025-08-07

Dear Reviewer Hu4h,

Thanks for your review! As the deadline is approaching, could you please check if the rebuttal addresses your concern and/or need additional clarification?

Best,

AC

---

### Note · Authors · 2025-08-14

We thank the reviewers and the area chair for their time and constructive feedback, and we appreciate the reviewers’ recognition of both the significance of the problem and the novelty of our proposed C-LoRA. As the rebuttal period concludes, we summarize the key points clarifying the contributions and novelty of our work.

C-LoRA introduces a principled framework for capturing input-dependent (aleatoric) uncertainty in large language model adaptation via contextualized low-rank adapters. By making the adaptation layers input-aware, C-LoRA models uncertainty tied to the semantics of each example without placing distributions over all model weights. This design enables meaningful stochasticity in predictions while keeping computational cost low and maintaining training stability. Across six common-sense reasoning benchmarks—covering both in-distribution and out-of-distribution datasets—C-LoRA demonstrates improved uncertainty quantification and calibration.

During the rebuttal, we clarified how C-LoRA’s contextualized adaptation layers capture input-dependent uncertainty and highlighted its conceptual and architectural differences from prior approaches. We also addressed questions about predictive variance behavior, evaluation scope, and design choices, showing that these components work together to yield the observed predictive performance and the improved uncertainty estimation.

In summary, C-LoRA provides a lightweight and principled approach for modeling aleatoric uncertainty in model adaptation. Across our evaluations, it achieves predictive performance comparable to existing methods while consistently improving uncertainty quantification and calibration. With the concerns addressed and the methodology and results clearly substantiated, we believe this work makes a meaningful contribution to uncertainty-aware model adaptation and will be of interest to the NeurIPS community.

---

### Decision · Program_Chairs · 2025-09-17

**Decision:**

Accept (poster)

**Comment:**

The paper introduces C-LORA (Contextual Low-Rank Adaptation), a parameter-efficient fine-tuning method for large language models (LLMs) that enhances uncertainty quantification by incorporating data-dependent contextual modules. Unlike traditional LoRA-based Bayesian approaches, which rely on static parameter posteriors, C-LORA dynamically adapts its uncertainty estimates based on each input. It achieves this through a small auxiliary network that predicts the distribution over an intermediate matrix inserted into the LoRA adapters, enabling explicit modeling of aleatoric uncertainty. The method is validated on six commonsense reasoning datasets, showing improved calibration (ECE, NLL) and robustness to distribution shift while maintaining competitive accuracy.

**Strengths:**
- Introduces a novel contextual mechanism that explicitly models aleatoric (input-specific) uncertainty, a significant step beyond prior Bayesian PEFT methods.
- Demonstrates consistent improvements in uncertainty calibration metrics (ECE, NLL) across multiple datasets and under distribution shift.
- Maintains computational efficiency by using a fixed-size latent representation and a compact context module, independent of model scale.
- Comprehensive experimental validation, including strong ablation studies confirming the effectiveness of the contextual module.
- Clear organization, detailed mathematical exposition, and plans to release code support reproducibility and clarity.

**Weaknesses:**
- Experiments are limited to Llama2-7B; no evaluation on larger models or other families (e.g., Mistral, Llama3), reducing generalizability.
- Slight decrease in task accuracy (1–2%) in certain datasets when compared to the best-performing baselines.
- The distinction between aleatoric and epistemic uncertainty is underdeveloped and may lead to conceptual ambiguity.
- Lacks qualitative illustrations of input-dependent uncertainty (e.g., visualizations or example-based variance analysis).
- Some notational clarity and architectural details (e.g., contextual module structure and training process) could be improved.

Most concerns have been addressed by the authors during the rebuttal period, and some reviewers have raised their scores accordingly, with 4 positive ratings at the end of the rebuttal. I am recommending acceptance. Note that

- One reviewer also mentioned that part of the proposed method uses MAP and is therefore not fully Bayesian, which I tend to agree. It is expected that the authors will make this clearer in the final version.
- The authors are expected to denote the fact that their method has only undergone testing on 7B as a limitation (preferably also mention this in the abstract) and that studying scaling of the underlying methodology remains an open problem.